# Efficient and Scalable Graph Generation through Iterative Local Expansion

**Andreas Bergmeister**[*]       **Karolis Martinkus**[†]       **Nathanaël Perraudin**[†]       **Roger Wattenhofer**
ETH Zürich            Prescient Design        SDSC, ETH Zürich        DISCO, ETH Zürich

## Abstract

In the realm of generative models for graphs, extensive research has been conducted. However, most existing methods struggle with large graphs due to the complexity of representing the entire joint distribution across all node pairs and capturing both global and local graph structures simultaneously. To overcome these issues, we introduce a method that generates a graph by progressively expanding a single node to a target graph. In each step, nodes and edges are added in a localized manner through denoising diffusion, building first the global structure, and then refining the local details. The local generation avoids modeling the entire joint distribution over all node pairs, achieving substantial computational savings with subquadratic runtime relative to node count while maintaining high expressivity through multiscale generation. Our experiments show that our model achieves state-of-the-art performance on well-established benchmark datasets while successfully scaling to graphs with at least 5000 nodes. Our method is also the first to successfully extrapolate to graphs outside of the training distribution, showcasing a much better generalization capability over existing methods.

## 1 Introduction

Graphs are mathematical structures representing relational data. They comprise a set of nodes and a set of edges, denoting pairwise relations between them. This abstraction is ubiquitous in modeling discrete data across domains like social networking (Fan et al., 2019), program synthesis (Nguyen et al., 2012; Bieber et al., 2020), or even origami design (Geiger et al., 2023). A crucial task is the generation of new graphs that possess characteristics similar to those observed. For example, in drug discovery, this involves creating graphs that encode the structure of a desired type of protein (Ingraham et al., 2022; Martinkus et al., 2023) or molecule (Jin et al., 2018; Vignac et al., 2023b).

Traditional graph generation methods (Albert & Barabási, 2002) estimate parameters of known models like Stochastic Block Model (SBM) (Holland et al., 1983) or Erdos-Renyi Erdos et al. (1960), but often fail to capture the complexity of real-world data. Deep learning offers a promising alternative, with approaches falling into two categories depending on the factorization of the data-generating distribution. Autoregressive techniques build graphs incrementally, predicting edges for each new node (You et al., 2018; Liao et al., 2020; Dai et al., 2020). One-shot methods generate the entire graph at once using techniques such as variational autoencoders (Simonovsky & Komodakis, 2018), generative adversarial networks (Cao & Kipf, 2022; Martinkus et al., 2022), normalizing flows (Liu et al., 2019), score-based and denoising diffusion models (Niu et al., 2020; Haefeli et al., 2023).

Despite the success of these methods in generating graphs comprising several hundred nodes, scaling beyond this range poses challenges. The computational cost of predicting edges between all node pairs scales at least quadratically with the number of nodes, which is inefficient for sparse graphs typical of real-world data. Sample fidelity is also an issue, as autoregressive methods struggle with node-permutation-invariant training due to the factorial increase in node orderings, and one-shot methods often fail to capture both global and local graph structure simultaneously. Also, in contrast to algorithmic approaches (Babiac et al., 2023), neither have been shown to generalize to larger unseen graph sizes. Finally, the neural architectures employed either exhibit limited expressiveness,

---

[*]Correspondence to: `andreas.bergmeister@inf.ethz.ch`
[†]Equal contribution.

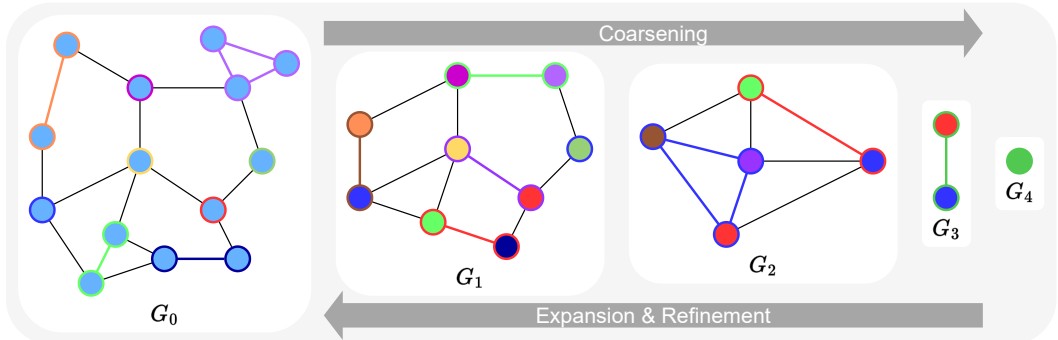

Figure 1: Example of a 4-level coarsening sequence. Colors indicate the node contraction sets $\mathcal{V}^{(p)}$. Our generation process aims at reversing with expansions and refinements the $T$ steps of this sequence from $G_T$ to $G_0$. The details of a single step are provided in Figure 2.

although with linear complexity in the number of edges for message passing neural networks (Xu et al., 2019), or are computationally expensive with quadratic or even higher scaling factors for more expressive architectures (Dwivedi & Bresson, 2021; Maron et al., 2020).

We present a novel approach to graph generation through iterative local expansion. In each step, we expand some nodes into small subgraphs and use a diffusion model to recover the appropriate local structure. The model is trained to reverse a graph coarsening process, as depicted in Figure 1, applied to the dataset graphs (Loukas, 2018; Loukas & Vandergheynst, 2018; Hermsdorff & Gunderson, 2019; Jin et al., 2020b; Kumar et al., 2022; 2023). We argue that this is inherently suitable for generating graphs, as it allows for the generation of an approximate global structure initially, followed by the addition of local details. This generative process effectively represents a particular kind of network growth, which we find to be much more robust to changes in generated graph sizes than existing approaches. Moreover, our method enables modeling the distribution of edges without the need to represent the entire joint distribution over all node pairs, enhancing scalability for larger graphs. Our theoretical analysis shows that, under mild conditions, our method exhibits subquadratic sampling complexity relative to the number of nodes for sparse graphs. We also introduce a more efficient local version of the Provably Powerful Graph Network (PPGN) (Maron et al., 2020), termed *Local PPGN*. This variant is especially well suited for our iterative local expansion approach, maintaining the high expressive power in the local subgraphs that we process while providing better computational efficiency. To demonstrate the effectiveness of our approach, we conducted experiments with widely used benchmark datasets. First, in the standard graph distribution modeling task from Martinkus et al. (2022); Vignac et al. (2023a), our model achieves state-of-the-art performance with the highest Validity-Uniqueness-Novelty score on the planar and tree datasets. Additionally, it generated graphs most closely matching the test set's structural statistics for protein and point cloud datasets. Second, we evaluate our method's ability to generalize beyond the training distribution, by generating graphs with an unseen number of nodes and verifying if they retain the defining characteristics of the training data. In this setting, our method is the only one capable of preserving these characteristics across the considered datasets. Third, we show that for sparse graphs our model exhibits subquadratic sampling complexity relative to the number of nodes, and validate this empirically by generating planar graphs of increasing size. Our implementation is available[1].

## 2 RELATED WORK

The seminal work by You et al. (2018) pioneered graph generation using recurrent neural networks, creating the adjacency matrix sequentially. Liao et al. (2020) improved this approach by simultaneously sampling edges for newly added nodes from a mixture of independent Bernoulli distributions with parameters obtained from a message passing graph neural network. Kong et al. (2023) conceptualized this method as the inverse of an absorbing state diffusion process (Austin et al., 2021) and proposed reinforcement learning to optimize node addition sequences.

Lately, diffusion models have come to dominate alternative approaches in terms of sample quality and diversity. Although initially only effective for graphs with tens of nodes (Niu et al., 2020), subsequent improvements using discrete diffusion (Vignac et al., 2023a;b; Haefeli et al., 2023), refining the diffusion process with a destination-predicting diffusion mixture (Jo et al., 2023), or dropping

---

[1]https://github.com/AndreasBergmeister/graph-generation

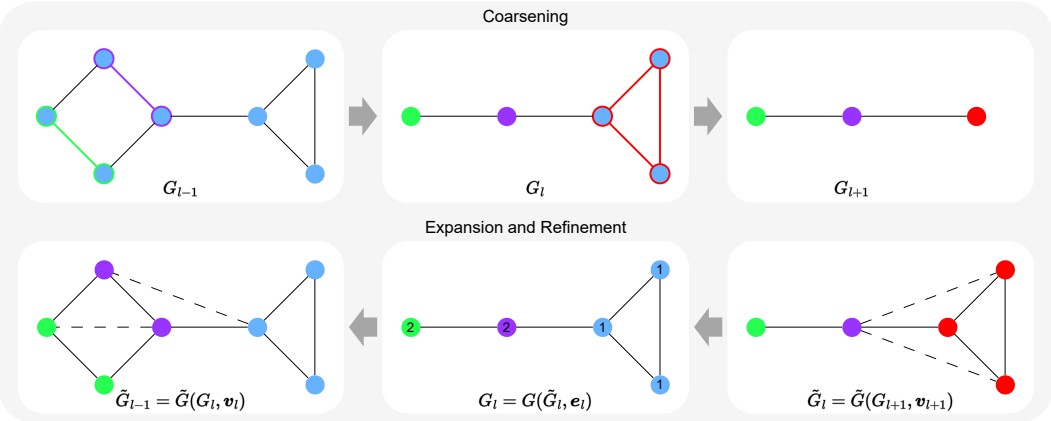

Figure 2: Single step schematic representation of the proposed methodology. The upper row delineates two sequential coarsening steps, using color differentiation to denote the contraction sets $\mathcal{V}^{(p)}$. Commencing from the right in the lower row, the expansion of $G_{l+1}$ into $\tilde{G}_l = \tilde{G}(G_{l+1}, \boldsymbol{v}_{l+1})$ is shown, assuming a known cluster size vector $\boldsymbol{v}_{l+1}$. Colors distinguish membership within expansion sets while dashed lines indicate edges to be removed as per the edge selection vector $\boldsymbol{e}_l$. The resultant refined graph $G_l = G(\tilde{G}_l, \boldsymbol{e}_l)$ is shown in the central box, where node features correspond to the cluster size vector $\boldsymbol{v}_l$, used in expanding $G_l$ into $\tilde{G}_{l-1}$ (illustrated in the leftmost box).

permutation equivariance (Yan et al., 2023) allowed for successful generation of graphs with a few hundred nodes. Nevertheless, scalability and computational complexity remain challenges for these models. As a countermeasure, Diamant et al. (2023) suggest limiting the maximal bandwidth of generated graphs. They leverage the observation that real-world graph nodes can often be ordered to confine non-zero adjacency matrix entries within a narrow diagonal band (Cuthill & McKee, 1969). Within this band, generation can be achieved using models such as GraphRNN (You et al., 2018), variational autoencoders (Grover et al., 2019), or score-based generative models (Niu et al., 2020). Alternatively, Chen et al. (2023b) introduce degree-guided diffusion, which begins with an RNN-generated degree sequence to condition the graph diffusion model. During each step, the model only considers edge connections between nodes predicted to require degree increases. This non-local process requires a simple, non-expressive, message passing graph neural network for efficient execution. However, it does offer an increase in empirical computational efficiency. Goyal et al. (2020) propose a different approach by generating a canonical string representation of the graph using a long short-term memory network. Although the length of the string is linear in the number of graph edges, generating the strings for model training has worst-case factorial complexity, which limits the practicality of this approach for general large-scale graph generation tasks.

An orthogonal line of research leverages hierarchical constructions for more efficient graph generation. Dai et al. (2020) improve the original RNN-based adjacency generation by You et al. (2018) using binary tree-structured conditioning on rows and columns of the matrix, cutting the complexity from $\mathcal{O}(n^2)$ to $\mathcal{O}((n + m)\log n)$, with $n$ representing nodes and $m$ edges. Shirzad et al. (2022) suggest a two-stage process starting with tree-based cluster representation, followed by incremental subgraph construction. Another two-level approach to generation is proposed by Davies et al. (2023) using DiGress (Vignac et al., 2023a) to create cluster graphs, followed by independent generation of cluster subgraphs and intra-cluster edges. In a related vein, Karami (2023) present a methodology that extends to multiple levels of hierarchy, with autoregressive generation of cluster subgraphs. Limnios et al. (2023) propose another method to enhance DiGress's scalability, which involves a divide-and-conquer strategy for sampling subgraph coverings. Although the independence assumptions of these hierarchical methods improve scalability, they may compromise sample accuracy, contrasting with our approach that avoids such assumptions. Both Davies et al. (2023) and Karami (2023) utilize the Louvain algorithm (Blondel et al., 2008) for pre-generating clusterings for training, unlike our method, which employs random sampling of coarsening sequences during training. Additionally, Guo et al. (2023) introduce a graph expansion layer for inclusion in the generator of a generative adversarial network or the decoder of a variational autoencoder, with parameter training carried out through reinforcement learning. Hierarchical approaches have also been developed for molecular generation (Jin et al., 2018; 2020a; Kuznetsov & Polykovskiy, 2021), with the aim of improving efficiency and performance by integrating domain knowledge. However, these methods are not optimized for general graph generation tasks.

## 3 METHOD

This section presents our proposed method for graph generation through iterative local graph expansion. A graph is a tuple $G = (\mathcal{V}, \mathcal{E})$, where $\mathcal{V}$ is a set of $n = |\mathcal{V}|$ vertices and $\mathcal{E}$ a set of $m = |\mathcal{E}|$ undirected edges. Assuming an arbitrary indexing of the nodes from 1 to $n$, we use $v^{(i)}$ to denote the $i$-th node in $\mathcal{V}$ and $e^{\{i,j\}} = \{v^{(i)}, v^{(j)}\} \in \mathcal{E}$ to denote the undirected edge connecting the nodes $v^{(i)}$ and $v^{(j)}$. Although the generated graphs are unattributed, the proposed method internally generates node and edge features denoted by $\boldsymbol{v}$ and $\boldsymbol{e}$ respectively. Their $i$-th component, denoted by $\boldsymbol{v}[i]$ and $\boldsymbol{e}[i]$, corresponds to the feature of the $i$-th node or edge in the graph. $\boldsymbol{W} \in \mathbb{R}^{n \times n}$ is a symmetric adjacency matrix with non-zero entries $\boldsymbol{W}[i,j] = \boldsymbol{W}[j,i]$ assigning positive (unary for the dataset graphs) weight to edges $e^{\{i,j\}} \in \mathcal{E}$. Consequently, the combinatorial Laplacian matrix is defined as $\boldsymbol{L} = \boldsymbol{D} - \boldsymbol{W}$, where $\boldsymbol{D}$ is the diagonal degree matrix with $\boldsymbol{D}[i,i] = \sum_{j=1}^{n} \boldsymbol{W}[i,j]$. All graphs are assumed to be connected.

### 3.1 GRAPH EXPANSION

Starting from a singleton graph $G_L = (\{v\}, \emptyset)$, we construct a sequence of graphs with increasing size in an auto-regressive fashion as

$$G_l \xrightarrow{\text{expand}} \tilde{G}_{l-1} \xrightarrow{\text{refine}} G_{l-1},$$

with $G_0$ being the graph to be generated. In every step, we expand each node in $G_l$ into a cluster of nodes, connecting nodes within the same cluster and between neighboring clusters, resulting in a graph $\tilde{G}_{l-1}$ with $n_{l-1}$ nodes. Subsequently, we refine $\tilde{G}_{l-1}$ into $G_{l-1}$ by selectively eliminating certain edges present in $\tilde{G}_{l-1}$. Figure 2 illustrates this process. Let us now formalize the definitions of the expansion and refinement steps.

**Definition 1 (Graph Expansion)** *Given a graph $G = (\mathcal{V}, \mathcal{E})$ with $|\mathcal{V}| = n$ nodes and a cluster size vector $\boldsymbol{v} \in \mathbb{N}^n$ denoting the expansion size of each node, let $\tilde{G}(G, \boldsymbol{v}) = (\tilde{\mathcal{V}}, \tilde{\mathcal{E}})$ denote the expansion of $G$. It contains $\boldsymbol{v}[p]$ nodes, $v^{(p_1)}, \ldots, v^{(p_{\boldsymbol{v}[p]})}$, for each node $v^{(p)} \in \mathcal{V}$ in the initial graph. As such, the expanded node set is given by $\tilde{\mathcal{V}} = \mathcal{V}^{(1)} \cup \cdots \cup \mathcal{V}^{(n)}$, where $\mathcal{V}^{(p)} = \{v^{(p_i)} \mid 1 \le i \le \boldsymbol{v}[p]\}$ for $1 \le p \le n$. The edge set $\tilde{\mathcal{E}}$ includes all intracluster edges, $\{e^{\{p_i, p_j\}} \mid 1 \le p \le n, 1 \le i < j \le \boldsymbol{v}[p]\}$, as well as the cluster interconnecting edges, $\{e^{\{p_i, q_j\}} \mid e^{\{p,q\}} \in \mathcal{E}, v^{(p_i)} \in \mathcal{V}^{(p)}, v^{(q_j)} \in \mathcal{V}^{(q)}\}$.*

**Definition 2 (Graph Refinement)** *Given a graph $\tilde{G} = (\tilde{\mathcal{V}}, \tilde{\mathcal{E}})$ with $\tilde{m} = \left|\tilde{\mathcal{E}}\right|$ edges and an edge selection vector $\boldsymbol{e} \in \{0,1\}^{\tilde{m}}$, let $G(\tilde{G}, \boldsymbol{e}) = (\mathcal{V}, \mathcal{E})$ denote the refinement of $\tilde{G}$, with $\mathcal{V} = \tilde{\mathcal{V}}$ and $\mathcal{E} \subseteq \tilde{\mathcal{E}}$ such that the $i$-th edge $e^{(i)} \in \mathcal{E}$ if and only if $\boldsymbol{e}[i] = 1$.*

**Probabilistic Model** Starting from a given dataset $\{G^{(1)}, \ldots, G^{(N)}\}$ of i.i.d. graph samples, we aim to fit a distribution $p(G)$ that matches the unknown true generative process as closely as possible. We model the marginal likelihood of a graph $G$ as the sum of likelihoods over expansion sequences

$$p(G) = \sum_{\varpi \in \Pi(G)} p(\varpi).$$

Here, $\Pi(G)$ denotes the set of all possible expansion sequences $(G_L = (\{v\}, \emptyset), G_{L-1}, \ldots, G_0 = G)$ of a single node into the target graph $G$, with each $G_{l-1}$ being a refined expansion of its predecessor, that is, $\tilde{G}_{l-1} = \tilde{G}(G_l, \boldsymbol{v}_l)$ is the expansion of $G_l$ according to Definition 1 with the cluster size vector $\boldsymbol{v}_l$, and $G_{l-1} = G(\tilde{G}_{l-1}, \boldsymbol{e}_{l-1})$ is the refinement of $\tilde{G}_{l-1}$ according to Definition 2 and the edge selection vector $\boldsymbol{e}_{l-1}$.

**Factorization** We factorize the likelihood of a fixed expansion sequence $\varpi = (G_L, \ldots, G_0)$ into a product of conditional likelihoods of single expansion and refinement steps, assuming a Markovian structure, as

$$p(\varpi) = \underbrace{p(G_L)}_{1} \cdot \prod_{l=L}^{1} p(G_{l-1} \mid G_l) = \prod_{l=L}^{1} p(\boldsymbol{e}_{l-1} \mid \tilde{G}_{l-1}) p(\boldsymbol{v}_l \mid G_l).$$

To avoid modeling two separate distributions $p(\boldsymbol{e}_l \mid \tilde{G}_l)$ and $p(\boldsymbol{v}_l \mid G_l)$, we rearrange terms as

$$p(\varpi) = \underbrace{p(\boldsymbol{v}_L \mid G_L)}_{p(\boldsymbol{v}_L)} \cdot \left[ \prod_{l=L-1}^{1} p(\boldsymbol{v}_l \mid G_l) p(\boldsymbol{e}_l \mid \tilde{G}_l) \right] \cdot p(\boldsymbol{e}_0 \mid \tilde{G}_0), \tag{1}$$

and model $\boldsymbol{v}_l$ to be conditionally independent of $\tilde{G}_l$ given $G_l$, i.e. $p(\boldsymbol{v}_l \mid G_l, \tilde{G}_l) = p(\boldsymbol{v}_l \mid G_l)$, allowing us to write

$$p(\boldsymbol{v}_l \mid G_l) p(\boldsymbol{e}_l \mid \tilde{G}_l) = p(\boldsymbol{v}_l, \boldsymbol{e}_l \mid \tilde{G}_l).$$

We represent the expansion and refinement vectors as node and edge features of the expanded graph, respectively. This enables us to model a single joint distribution over these features for each refinement and consecutive expansion step.

## 3.2 LEARNING TO INVERT GRAPH COARSENING

We now describe how we construct expansion sequences $\varpi \in \Pi(G)$ for a given graph $G$ and use them to train a model for conditional distributions $p(\boldsymbol{v}_l, \boldsymbol{e}_l \mid \tilde{G}_l)$. For this, we introduce the notion of graph coarsening as the inverse operation of graph expansion. Intuitively, we obtain a coarsening of a graph by partitioning its nodes into nonoverlapping, connected sets and contracting the induced subgraph of each set into a single node.

**Definition 3 (Graph Coarsening)** *Let $G = (\mathcal{V}, \mathcal{E})$ be an arbitrary graph and $\mathcal{P} = \{\mathcal{V}^{(1)}, \ldots, \mathcal{V}^{(\bar{n})}\}$ be a partitioning of the node set $\mathcal{V}$, such that each partition $\mathcal{V}^{(p)} \in \mathcal{P}$ induces a connected subgraph in $G$. We construct a coarsening $\bar{G}(G, \mathcal{P}) = (\bar{\mathcal{V}}, \bar{\mathcal{E}})$ of $G$ by representing each partition $\mathcal{V}^{(p)} \in \mathcal{P}$ as a single node $v^{(p)} \in \bar{\mathcal{V}}$. We add an edge $e^{\{p,q\}} \in \bar{\mathcal{E}}$, between distinct nodes $v^{(p)} \neq v^{(q)} \in \bar{\mathcal{V}}$ in the coarsened graph if and only if there exists an edge $e^{\{i,j\}} \in \mathcal{E}$ between the corresponding disjoint clusters in the original graph, i.e. $v^{(i)} \in \mathcal{V}^{(p)}$ and $v^{(j)} \in \mathcal{V}^{(q)}$.*

An important property of this coarsening operation is that it can be inverted through an appropriate expansion and subsequent refinement step, as elaborated in Appendix A. Based on this premise, it can be deduced through an inductive argument that for any given coarsening sequence $(G = G_0, G_1, \ldots, G_L = (\{v\}, \emptyset))$ that transforms a graph $G$ into a single node, there exists a corresponding expansion sequence $\varpi \in \Pi(G)$ with the same elements in reverse order, i.e. $\varpi = (G_L, \ldots, G_0)$. Note that successive coarsening steps always result in a single-node graph, as long as the original graph is connected, and every coarsening step contains at least one non-trivial contraction set, i.e. a set of nodes with more than one node.

We define the distribution $p(\pi)$ over coarsening sequences symmetrically to $p(\varpi)$ in Equation 1 and use $\Pi(G)$ to denote the set of all possible coarsening sequences of a graph $G$. With this, it holds that

$$p(G) = \sum_{\varpi \in \Pi(G)} p(\varpi) \geq \sum_{\pi \in \Pi(G)} p(\pi). \tag{2}$$

Note that this inequality is strict, as there exist expansion sequences that are not the reverse of any coarsening sequence[2]. As we can easily generate samples from $\Pi(G)$, this is a suitable lower bound on the marginal likelihood of $G$ that we can aim to maximize during training.

**Contraction Families** Without further restrictions on the allowed partitioning of the node set in Definition 3, for an arbitrary graph $G$ there can potentially be exponentially many coarsenings of it, rendering the computation of the sum $\sum_{\pi \in \Pi(G)} p(\pi)$ intractable. Therefore, we further restrict the possible contraction sets in graph coarsening to belong to a given contraction family $\mathcal{F}(G)$. We use $\Pi_{\mathcal{F}}(G)$ to denote the set of all possible coarsening sequences of $G$ that only use contraction sets from $\mathcal{F}(G)$ in each step. $\Pi_{\mathcal{F}}(G)$ is a subset of $\Pi(G)$, and hence Equation 2 with $\Pi(G)$ replaced by $\Pi_{\mathcal{F}}(G)$ still holds. Following Loukas (2018), we experiment with edge contraction $\mathcal{F}(G) = \mathcal{E}$ and neighborhood contraction $\mathcal{F}(G) = \{\{v^{(j)} \mid e^{\{i,j\}} \in \mathcal{E}\} \mid v^{(i)} \in \mathcal{V}\}$.

---

[2]For example, the refinement step might split the graph into two connected components, which cannot, from Definition 3, be coarsened back into a single connected graph.

**Variational Interpretation**   Given a distribution $q(\pi \mid G)$ over coarsening sequences $\Pi_{\mathcal{F}}(G)$ for a graph $G$, it holds that

$$p(G) \geq \sum_{\pi \in \Pi_{\mathcal{F}}(G)} p(\pi) \geq \mathop{\mathbb{E}}_{\pi \sim q(\pi \mid G)} \left[ \frac{p(\pi)}{q(\pi \mid G)} \right],$$

and one can derive the evidence lower bound on the log-likelihood under the given model as

$$\log p(G) \geq \mathop{\mathbb{E}}_{\pi \sim q(\pi \mid G)} \left[ \log p(\boldsymbol{v}_L \mid G_L) + \sum_{l=L-1}^{1} \log p(\boldsymbol{v}_l, \boldsymbol{e}_l \mid \tilde{G}_l) + \log p(\boldsymbol{e}_0 \mid \tilde{G}_0) \right] + \mathbb{H}(q(\pi \mid G)), \tag{3}$$

leading to a variational interpretation of the model.

**Spectral Guided Generation**   The above formulation is agnostic to the distribution $q(\pi \mid G)$ over coarsening sequences $\Pi_{\mathcal{F}}(G)$, giving us the flexibility to choose a distribution that facilitates the learning process and improves the generative performance of the model. While the uniform distribution over all possible coarsening sequences $\Pi_{\mathcal{F}}(G)$ gives the tightest bound in Equation 3 as the entropy term vanishes, arbitrary coarsening sequences could destroy important structural properties of the original graph $G$, making it difficult for the model to learn to invert them. Therefore, we propose a distribution $q$ that prioritizes coarsening sequences preserving the spectrum of the graph Laplacian, which is known to capture important structural properties of a graph. Note that the distribution $q$ does not need to be explicitly defined. Instead, for training the model, we only need a sampling procedure from this distribution. In Appendix D, we propose a sampling procedure for coarsening sequences which is parametric in a cost function. It iteratively evaluates the cost function across all contraction sets and subsequently selects a cost-minimizing partition of the contraction sets in a greedy and stochastic fashion. When instantiating the cost function with the *Local Variation Cost* 7 proposed by Loukas (2018), we obtain a Laplacian spectrum-preserving distribution over coarsening sequences. In Appendix C, we summarize the work of Loukas (2018) and show how our generic sampling procedure can be instantiated with the *Local Variation Cost*. In Section D.1, we empirically validate the effectiveness of this approach by comparing the generative performance of the model with and without spectrum-preserving sampling. While numerous graph coarsening techniques exist (Loukas, 2018; Hermsdorff & Gunderson, 2019; Jin et al., 2020b; Kumar et al., 2022; 2023), our chosen method stands out for two key reasons. It adheres to our coarsening definition with an efficient local cost function guiding contraction set selection. Additionally, it's a multilevel scheme that maintains the original graph's Laplacian spectrum at each level, essential for our goals.

### 3.3   MODELING AND TRAINING

We now turn to the modeling of conditional distributions $p(\boldsymbol{v}_l, \boldsymbol{e}_l \mid \tilde{G}_l)$ within our marginal likelihood factorization for $p(G)$. Let $p_{\boldsymbol{\theta}}(\boldsymbol{v}_l, \boldsymbol{e}_l \mid \tilde{G}_l)$ denote the parameterized distribution, with $\theta$ as the parameters. We use the same model for all $1 \leq l < L$ conditional distributions $p_{\boldsymbol{\theta}}(\boldsymbol{v}_l, \boldsymbol{e}_l \mid \tilde{G}_l)$ as well as $p_{\boldsymbol{\theta}}(\boldsymbol{e}_0 \mid \tilde{G}_0)$ and $p_{\boldsymbol{\theta}}(\boldsymbol{v}_L \mid G_L) = p_{\boldsymbol{\theta}}(\boldsymbol{v}_L)$, with the parameters $\boldsymbol{\theta}$ being shared between all distributions. For the latter two distributions, we disregard the edge and node features, respectively, but maintain the same modeling approach as for the other distributions. In the following, we describe the modeling of $p_{\boldsymbol{\theta}}(\boldsymbol{v}_l, \boldsymbol{e}_l \mid \tilde{G}_l)$ for arbitrary but fixed level $1 \leq l < L$.

**Modeling with Denoising Diffusion Models**   An effective method should be capable of representing complex distributions and provide a stable, node permutation-invariant training loss. Denoising diffusion models meet these criteria. This method entails training a denoising model to restore the original samples—in our setting, node and edge features $\boldsymbol{v}_l$ and $\boldsymbol{e}_l$—from their corrupted counterparts. Inference proceeds iteratively, refining predictions from an initial noise state. Although this requires multiple model queries per graph expansion, it does not affect the algorithm's asymptotic complexity nor impose restrictive assumptions on the distribution, unlike simpler models such as mixtures of independent categorical distributions. We adopt the formulation proposed by Song et al. (2021), enhanced by contributions from Karras et al. (2022). This method represents the forefront in image synthesis, and preliminary experiments indicate its superior performance for our application. For a comprehensive description of the framework and its adaptation to our context, see Appendix E.

### 3.4 LOCAL PPGN

A key component of our proposed methodology is the specialized architecture designed to parameterize the conditional distributions $p_{\theta}(\boldsymbol{v}_l, \boldsymbol{e}_l \mid \tilde{G}_l)$, or equivalently, the denoising model. Our design incorporates a novel edge-wise message passing layer, termed *Local PPGN*. When designing this layer, we drew inspiration from the *PPGN* model (Maron et al., 2020), which is provably more expressive than graph message passing networks at the expense of increased computational complexity (cubic in the number of nodes). Recognizing that our suggested methodology only locally alternates graphs at every expansion step and that these graphs possess a locally dense structure, as a result of the expansion process (Definition 1), we designed a layer that is locally expressive, resembles the *PPGN* layer on a dense (sub)graph, but retains efficiency on sparse graphs, with linear runtime relative to the number of edges. An elaborate explanation of this layer and its placement within existing graph neural network models can be found in Appendix F. In-depth architectural details of the overall model are presented in Appendix F.2.

### 3.5 SPECTRAL CONDITIONING

Martinkus et al. (2022) found that using the principal Laplacian eigenvalues and eigenvectors of a target graph as conditional information improves graph generative models. A salient aspect of our generative methodology is that it generates a graph $G_l$ from its coarser version, $G_{l+1}$. Given the preservation of the spectrum during coarsening, the Laplacian spectrum of $G_l$ is approximated by that of $G_{l+1}$. The availability of $G_{l+1}$ during the generation of $G_l$ allows computing its principal Laplacian spectrum and subsequently conditioning the generation of $G_l$ on it. Specifically, we accomplish this by computing the smallest $k$ non-zero eigenvalues and their respective eigenvectors of the Laplacian matrix $\boldsymbol{L}_{l+1}$ of $G_{l+1}$. We then employ *SignNet* (Lim et al., 2022) to obtain node embeddings for nodes in $G_{l+1}$, which are then replicated across nodes in the same expansion set to initialize $G_l$'s embeddings. This shared embedding feature also aids the model in cluster identification. Our *Local PPGN* model, while inherently capturing global graph structures, can benefit from explicit conditioning on spectral information. We adjust the number of eigenvalues $k$ as a tunable hyperparameter; when $k = 0$, node embeddings are drawn from an isotropic normal distribution.

### 3.6 PERTURBED EXPANSION

As noted, the given Definitions 1 and 2 are sufficient to reverse a contraction step with an appropriate expansion and subsequent refinement step. However, we have observed that introducing an additional source of randomness in the expansion is beneficial for the generative performance of the model, particularly in the context of datasets with limited samples where overfitting is a concern. Therefore, we introduce the concept of perturbed expansion, where in addition to the edges in $\tilde{\mathcal{E}}$, we add edges between nodes whose distance in $G$ is bounded by an augmented radius independently with a given probability. A formal definition and an illustrative explanation of this concept can be found in Appendix B.

### 3.7 DETERMINISTIC EXPANSION SIZE

Our graph expansion method iteratively samples a cluster size vector $\boldsymbol{v}$ to incrementally enlarge the graph. The process halts when $\boldsymbol{v}$ is entirely composed of ones, indicating no further node expansion is necessary. However, this stochastic approach may not reliably produce graphs of a predetermined size. To remedy this, we propose a deterministic expansion strategy, primarily applicable in cases of edge contraction where the maximum expansion size is two. In this strategy, $\boldsymbol{v}$ is treated as binary. We set the target size for the expanded graph at each expansion step and, instead of sampling $\boldsymbol{v}$, we select the required number of nodes with the highest probabilities for expansion to reach the predefined size. Additionally, we introduce the reduction fraction, calculated as one minus the ratio of node counts between the original and expanded graphs, as an additional input to the model during training and inference. More details are discussed in Appendix G.

| | Planar Graphs ($n_{max} = 64$, $n_{avg} = 64$) | | | | | | | | | |
|---|---|---|---|---|---|---|---|---|---|---|
| Model | Deg.↓ | Clus.↓ | Orbit↓ | Spec.↓ | Wavelet↓ | Ratio↓ | Valid↑ | Unique↑ | Novel↑ | V.U.N.↑ |
| Training set | 0.0002 | 0.0310 | 0.0005 | 0.0038 | 0.0012 | 1.0 | 100 | 100 | — | — |
| GraphRNN (You et al., 2018) | 0.0049 | 0.2779 | 1.2543 | 0.0459 | 0.1034 | 490.2 | 0.0 | 100 | 100 | 0.0 |
| GRAN (Liao et al., 2020) | 0.0007 | 0.0426 | 0.0009 | 0.0075 | 0.0019 | 2.0 | 97.5 | 85.0 | 2.5 | 0.0 |
| SPECTRE (Martinkus et al., 2022) | 0.0005 | 0.0785 | 0.0012 | 0.0112 | 0.0059 | 3.0 | 25.0 | 100 | 100 | 25.0 |
| DiGress (Vignac et al., 2023a) | 0.0007 | 0.0780 | 0.0079 | 0.0098 | 0.0031 | 5.1 | 77.5 | 100 | 100 | 77.5 |
| EDGE (Chen et al., 2023b) | 0.0761 | 0.3229 | 0.7737 | 0.0957 | 0.3627 | 431.4 | 0.0 | 100 | 100 | 0.0 |
| BwR (EDP-GNN) (Diamant et al., 2023) | 0.0231 | 0.2596 | 0.5473 | 0.0444 | 0.1314 | 251.9 | 0.0 | 100 | 100 | 0.0 |
| BiGG (Dai et al., 2020) | 0.0007 | 0.0570 | 0.0367 | 0.0105 | 0.0052 | 16.0 | 62.5 | 85.0 | 42.5 | 5.0 |
| GraphGen (Goyal et al., 2020) | 0.0328 | 0.2106 | 0.4236 | 0.0430 | 0.0989 | 210.3 | 7.5 | 100 | 100 | 100 |
| Ours (one-shot) | 0.0003 | 0.0245 | 0.0006 | 0.0104 | 0.0030 | **1.7** | 67.5 | 100 | 100 | 67.5 |
| Ours | 0.0005 | 0.0626 | 0.0017 | 0.0075 | 0.0013 | 2.1 | 95.0 | 100 | 100 | **95.0** |
| | Stochastic Block Model ($n_{max} = 187$, $n_{avg} = 104$) | | | | | | | | | |
| Model | Deg.↓ | Clus.↓ | Orbit↓ | Spec.↓ | Wavelet↓ | Ratio↓ | Valid↑ | Unique↑ | Novel↑ | V.U.N.↑ |
| Training set | 0.0008 | 0.0332 | 0.0255 | 0.0027 | 0.0007 | 1.0 | 100 | 100 | — | — |
| GraphRNN (You et al., 2018) | 0.0055 | 0.0584 | 0.0785 | 0.0065 | 0.0431 | 14.7 | 5.0 | 100 | 100 | 5.0 |
| GRAN (Liao et al., 2020) | 0.0113 | 0.0553 | 0.0540 | 0.0054 | 0.0212 | 9.7 | 25.0 | 100 | 100 | 25.0 |
| SPECTRE (Martinkus et al., 2022) | 0.0015 | 0.0521 | 0.0412 | 0.0056 | 0.0028 | 2.2 | 52.5 | 100 | 100 | 52.5 |
| DiGress (Vignac et al., 2023a) | 0.0018 | 0.0485 | 0.0415 | 0.0045 | 0.0014 | **1.7** | 60.0 | 100 | 100 | 60.0 |
| EDGE (Chen et al., 2023b) | 0.0279 | 0.1113 | 0.0854 | 0.0251 | 0.1500 | 51.4 | 0.0 | 100 | 100 | 0.0 |
| BwR (EDP-GNN) (Diamant et al., 2023) | 0.0478 | 0.0638 | 0.1139 | 0.0169 | 0.0894 | 38.6 | 7.5 | 100 | 100 | 7.5 |
| BiGG (Dai et al., 2020) | 0.0012 | 0.0604 | 0.0667 | 0.0059 | 0.0370 | 11.9 | 10.0 | 100 | 100 | 10.0 |
| GraphGen (Goyal et al., 2020) | 0.0550 | 0.0623 | 0.1189 | 0.0182 | 0.1193 | 48.8 | 5.0 | 100 | 100 | 5.0 |
| Ours (one-shot) | 0.0141 | 0.0528 | 0.0809 | 0.0071 | 0.0205 | 10.5 | 75.0 | 100 | 100 | **75.0** |
| Ours | 0.0119 | 0.0517 | 0.0669 | 0.0067 | 0.0219 | 10.2 | 45.0 | 100 | 100 | 45.0 |
| | Tree Graphs ($n_{max} = 64$, $n_{avg} = 64$) | | | | | | | | | |
| Model | Deg.↓ | Clus.↓ | Orbit↓ | Spec.↓ | Wavelet↓ | Ratio↓ | Valid↑ | Unique↑ | Novel↑ | V.U.N.↑ |
| Training set | 0.0001 | 0.0000 | 0.0000 | 0.0075 | 0.0030 | 1.0 | 100 | 100 | — | — |
| GRAN (Liao et al., 2020) | 0.1884 | 0.0080 | 0.0199 | 0.2751 | 0.3274 | 607.0 | 0.0 | 100 | 100 | 0.0 |
| DiGress (Vignac et al., 2023a) | 0.0002 | 0.0000 | 0.0000 | 0.0113 | 0.0043 | **1.6** | 90.0 | 100 | 100 | 90.0 |
| EDGE (Chen et al., 2023b) | 0.2678 | 0.0000 | 0.7357 | 0.2247 | 0.4230 | 850.7 | 0.0 | 7.5 | 100 | 0.0 |
| BwR (EDP-GNN) (Diamant et al., 2023) | 0.0016 | 0.1239 | 0.0003 | 0.0480 | 0.0388 | 11.4 | 0.0 | 100 | 100 | 0.0 |
| BiGG (Dai et al., 2020) | 0.0014 | 0.0000 | 0.0000 | 0.0119 | 0.0058 | 5.2 | 100 | 87.5 | 50.0 | 75.0 |
| GraphGen (Goyal et al., 2020) | 0.0105 | 0.0000 | 0.0000 | 0.0153 | 0.0122 | 33.2 | 95.0 | 100 | 100 | 95.0 |
| Ours (one-shot) | 0.0004 | 0.0000 | 0.0000 | 0.0080 | 0.0055 | 2.1 | 82.5 | 100 | 100 | 82.5 |
| Ours | 0.0001 | 0.0000 | 0.0000 | 0.0117 | 0.0047 | 4.0 | 100 | 100 | 100 | **100** |

Table 1: Sample quality on synthetic graphs.

| | Proteins ($n_{max} = 500$, $n_{avg} = 258$) | | | | | | Point Clouds ($n_{max} = 5037$, $n_{avg} = 1332$) | | | | | |
|---|---|---|---|---|---|---|---|---|---|---|---|---|
| Model | Deg.↓ | Clus.↓ | Orbit↓ | Spec.↓ | Wavelet↓ | Ratio↓ | Deg.↓ | Clus.↓ | Orbit↓ | Spec.↓ | Wavelet↓ | Ratio↓ |
| Training set | 0.0003 | 0.0068 | 0.0032 | 0.0005 | 0.0003 | 1.0 | 0.0000 | 0.1768 | 0.0049 | 0.0043 | 0.0024 | 1.0 |
| GraphRNN (You et al., 2018) | 0.004 | 0.1475 | 0.5851 | 0.0152 | 0.0530 | 91.3 | OOM | OOM | OOM | OOM | OOM | OOM |
| GRAN (Liao et al., 2020) | 0.0479 | 0.1234 | 0.3458 | 0.0125 | 0.0341 | 87.5 | 0.0201 | 0.4330 | 0.2625 | 0.0051 | 0.0436 | 18.8 |
| SPECTRE (Martinkus et al., 2022) | 0.0056 | 0.0843 | 0.0267 | 0.0052 | 0.0118 | 19.0 | OOM | OOM | OOM | OOM | OOM | OOM |
| DiGress (Vignac et al., 2023a) | 0.0041 | 0.0489 | 0.1286 | 0.0018 | 0.0065 | 18.0 | OOM | OOM | OOM | OOM | OOM | OOM |
| EDGE (Chen et al., 2023b) | 0.1863 | 0.3406 | 0.6786 | 0.1075 | 0.2371 | 399.1 | 0.4441 | 0.3298 | 1.0730 | 0.4006 | 0.6310 | 143.4 |
| BwR (EDP-GNN) (Diamant et al., 2023) | 0.1262 | 0.4202 | 0.4939 | 0.0702 | 0.1199 | 245.4 | 0.4927 | 0.4690 | 1.0730 | 0.2912 | 0.5916 | 133.2 |
| BiGG (Dai et al., 2020) | 0.0070 | 0.1150 | 0.4696 | 0.0067 | 0.0222 | 57.5 | 0.0994 | 0.6035 | 0.3633 | 0.1589 | 0.0994 | 38.8 |
| GraphGen (Goyal et al., 2020) | 0.0159 | 0.1677 | 0.3789 | 0.0181 | 0.0477 | 83.5 | OOT | OOT | OOT | OOT | OOT | OOT |
| Ours (one-shot) | 0.0015 | 0.0711 | 0.0396 | 0.0026 | 0.0086 | 13.3 | OOM | OOM | OOM | OOM | OOM | OOM |
| Ours | 0.0030 | 0.0309 | 0.0047 | 0.0013 | 0.0030 | **5.9** | 0.0139 | 0.5775 | 0.0780 | 0.0055 | 0.0186 | **7.0** |

Table 2: Sample quality on real-world graphs. All models achieve perfect uniqueness and novelty. Several models fail on the point cloud dataset due to memory limitations (OOM), and GraphGen is unable to generate the canonical string representations within a reasonable timeframe (OOT).

## 4 EXPERIMENTS

Our experiments evaluate three main aspects of our model: (1) its ability to generate graphs with structural properties similar to the training data on common synthetic graph generation datasets (planar, SBM, tree); (2) its ability to scale to much larger real-world graphs (proteins and point clouds); (3) extrapolation to out-of-distribution graph sizes. We rely on the standard metrics, datasets and evaluation procedures introduced by Martinkus et al. (2022). Details on this and the hyperparameters we used are covered in Appendix I.

**Simple Graph Generation.** In Table 1 the most critical metric is the percentage of valid, unique, and novel graphs (V.U.N) in the generated set. Validity for synthetic graphs indicates the adherence to the defined properties, e.g. planarity or acyclicity. Uniqueness and novelty metrics report the diversity of the output, serving as an indicator for non-overfitting. Our method demonstrates strong performance, surpassing our baseline, which operates without iterative expansion, but directly generates the full graph using the diffusion model (Ours (one-shot)). The exception is the SBM dataset, where the inherent randomness of the graphs and the absence structure aside from large clusters, likely affects the results. Nevertheless, our model still attains a satisfactory V.U.N. score. The first

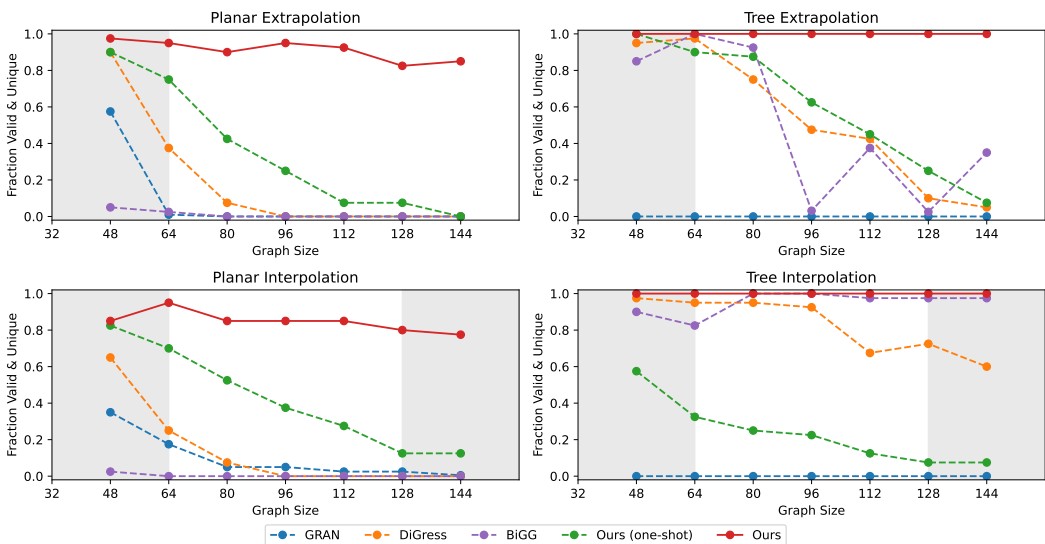

Figure 3: Extrapolation and interpolation to out-of-distribution graph sizes. The shaded area represents the training size range.

five columns of the table show the maximum mean discrepancy (MMD) between the generated and test graphs for the degree distribution, clustering coefficient, orbit counts, spectrum, and wavelet coefficients. We summarize these metrics by the average ratio between the generated and training MMDs. Although DiGress is the overall best performer with respect to this metric, our method achieves competitive results and is the best for planar graphs.

The benefits of our approach become clearer with larger, complex real-world graphs. In Table 2 we show model performance on protein graphs with up to 500 nodes and point cloud graphs with up to 5037 nodes (see Table 6). In both cases, our method outperforms competitors by a large margin in structural similarity to the test set. Note that several methods are unable to scale to 5037 nodes.

Appendix H offers a runtime comparison, affirming our method's subquadratic scaling when generating sparse graphs of increasing size. This section also includes a theoretical analysis of the model's complexity. Sample graphs generated by our model can be found in Appendix J.

**Extrapolation and Interpolation.** We assess our model's capability to generate graphs with node counts beyond the training distribution through extrapolation (creating larger graphs) and interpolation (varying sizes within observed ranges). We use a planar and a tree dataset, each comprising 128 training graphs with sizes uniformly sampled from $[32, 64]$ for extrapolation and from $[32, 64] \cup [128, 160]$ for interpolation. Our evaluation involves generating graphs with 48 to 144 nodes, producing 32 graphs per size for validation and 40 for testing. We report the validity and uniqueness rates of generated graphs.

Figure 3 demonstrates that our method is uniquely capable of reliably extrapolating and interpolating to out-of-distribution graph sizes across both datasets. We note that GRAN, DiGress and Ours (one shot) fail, in general, to generate larger graphs in contrast to their performance on smaller versions of the datasets (see Table 1). Therefore, our experiment does not fully determine whether these methods fail because they cannot interpolate/extrapolate or because they are unable to generate larger graphs.

## 5 CONCLUSION

In this work, we present the first graph generative method based on iterative local expansion, where generation is performed by a single model that iteratively expands a single node into the full graph. We made our method efficient (with sub-quadratic complexity) by introducing the *Local PPGN* layer that retains high expressiveness while performing only local computation. We performed tests on traditional graph generation benchmarks, where our method achieved state-of-the-art results. Furthermore, to the best of our knowledge, our method is the only one able to generate graphs outside of the training distribution (with different numbers of nodes) while retaining the main graph characteristics across different datasets.

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

## A  INVERT COARSENING BY EXPANSION AND REFINEMENT

In this Appendix, we show that each coarsen graph (according to Definition 3) can be inverted with a specific expansion and refinement step.

Let $G = (\mathcal{V}, \mathcal{E})$ be an arbitrary graph and $\mathcal{P}$ a connected subgraph that induces partitioning of its node set. Furthermore, let $G_c = (\mathcal{V}_c, \mathcal{E}_c) = \bar{G}(G, \mathcal{P})$ denote the coarsened graph according to Definition 3. In the following, we construct an expansion and refinement vector that recovers the original graph $G$ from its coarsening $G_c$.

We start with the expansion by setting the vector $\boldsymbol{v} \in \mathbb{N}^{|\mathcal{P}|}$ to

$$\boldsymbol{v}[p] = \left|\mathcal{V}^{(p)}\right| \quad \text{for all} \quad \mathcal{V}^{(p)} \in \mathcal{P}. \tag{4}$$

Let $G_e = (\mathcal{V}_e, \mathcal{E}_e) = \tilde{G}(G_c, \boldsymbol{v})$ denote the expanded graph as per Definition 1 (or Definition 4). It is important to note that the node sets of $G$ and $G_e$ possess the same cardinality. Thus, we can establish a bijection $\varphi : \mathcal{V} \to \mathcal{V}_e$ between them, where the $i$-th node in the $p$-th partition of $\mathcal{P}$ is mapped to the corresponding node $v^{(p_i)} \in \mathcal{V}_e$ within the expanded graph. This construction leads to the edge set of $G_e$ being a superset of the one of the original graph $G$. To see why, consider an arbitrary edge $e^{\{i,j\}} \in \mathcal{E}$. If both $v^{(i)}$ and $v^{(j)}$ belong to the same partition $\mathcal{V}^{(p)} \in \mathcal{P}$, they are contracted to a common node $v^{(p)}$ in $G_c$. When expanding $v^{(p)}$ to a set of $\left|\mathcal{V}^{(p)}\right|$ nodes in $G_e$, by construction of the intercluster edge set in Definition 1 (resp. Definition 4), all $\left|\mathcal{V}^{(p)}\right|$ nodes are connected in $G_e$. On the other hand, if $v^{(i)}$ and $v^{(j)}$ lie in different partitions $\mathcal{V}^{(p)}$ and $\mathcal{V}^{(q)}$, respectively, an edge in the original graph implies that the partitions representing the nodes $v^{(p)} \in \mathcal{V}_c$ and $v^{(q)} \in \mathcal{V}_c$ are connected in $G_c$. Consequently, when expanding $v^{(p)}$ and $v^{(q)}$, all $\left|\mathcal{V}^{(p)}\right|$ nodes associated with $v^{(p)}$ are connected to all $\left|\mathcal{V}^{(q)}\right|$ nodes associated with $v^{(q)}$ in $G_e$. Specifically, $v^{(\varphi(i))}$ and $v^{(\varphi(j))}$ are connected in $G_e$.

Therefore, for the refinement step, we define the vector $\boldsymbol{e} \in \mathbb{N}^{|\mathcal{E}_e|}$ as follows: given an arbitrary ordering of the edges in $\mathcal{E}_e$, let $e^{\{i,j\}} \in \mathcal{E}_e$ denote the $i$-th edge in this ordering, we set

$$\boldsymbol{e}[i] = \begin{cases} 1 & \text{if} \quad e^{\{\varphi^{-1}(i), \varphi^{-1}(j)\}} \in \mathcal{E} \\ 0 & \text{otherwise.} \end{cases} \tag{5}$$

As per Definition 2, the refined graph is then given by $G_r = \bar{G}(G_e, \boldsymbol{e}) = \bar{G}(\tilde{G}(G_c, \boldsymbol{v}), \boldsymbol{e})$, which is isomorphic to the original graph $G$.

## B PERTURBED GRAPH EXPANSION

The following definition formalizes the concept of a randomized graph expansion, which is a generalization of the deterministic expansion introduced in Definition 1. A visual representation of this concept is provided in Figure 4.

**Definition 4 (Perturbed Graph Expansion)** *Given a graph $G = (\mathcal{V}, \mathcal{E})$, a cluster size vector $\boldsymbol{v} \in \mathbb{N}^n$, a radius $r \in \mathbb{N}$, and a probability $0 \leq p \leq 1$, the perturbed expansion $\tilde{G}$ is constructed as in Definition 1, and additionally for all distinct nodes $v^{(p)}, v^{(q)} \in \tilde{\mathcal{V}}$ whose distance in $G$ is at most $r$, we add each edge $e^{\{p_i, q_j\}}$ independently to $\tilde{\mathcal{E}}$ with probability $p$.*

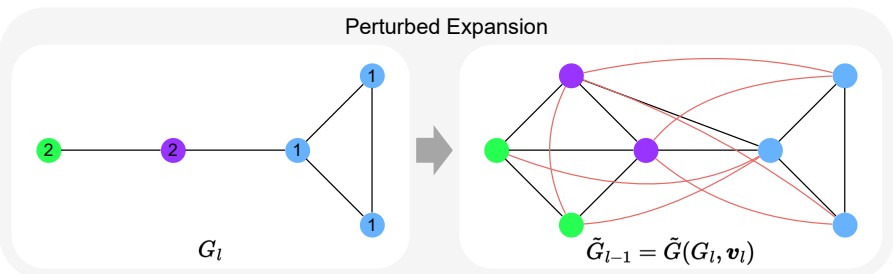

Figure 4: Depiction of a perturbed expansion. The graph $G_l$ is expanded into $\tilde{G}_{l-1}$ utilizing the cluster size vector aligned to the node features. Deterministic expansion components are represented by linear black edges, whereas curved red edges showcase supplemental edges implemented for a radius of $r = 2$ and a probability of $p = 1$. With $p < 1$, a subset of these edges would be randomly excluded.

## C  SPECTRUM-PRESERVING COARSENING

The work of Loukas (2018) presents a multi-level graph coarsening algorithm designed to reduce the size of a graph while ensuring that the Laplacian spectrum of the coarsened graph closely approximates the spectrum of the original graph. Since the Laplacian spectrum captures crucial structural properties of a graph, this implies that fundamental structural properties of the original graph, such as (normalized) cuts, are preserved by the coarsening. The proposed reduction scheme aligns well with our framework, as it defines graph coarsening in a similar manner to Definition 3, and the selection of subgraphs for contraction involves computing a cost for each contraction set.

Let us now summarize the key aspects of the proposed framework and illustrate how we adapt it to our setting. We consider a graph $G = (\mathcal{V}, \mathcal{E})$ with $n$ nodes and a Laplacian $\boldsymbol{L}$. Let $\mathcal{P} = \{\mathcal{V}^{(1)}, \dots, \mathcal{V}^{(n)}\}$ be a partition of the node set $\mathcal{V}$ into non-overlapping, connected sets, and $G_c = (\mathcal{V}_c, \mathcal{E}_c) = \bar{G}(G, \mathcal{P})$ the coarsened graph according to Definition 3 with $n_c$ nodes.

Comparing the Laplacian matrices of $G$ and $G_c$ requires us to relate signals $x \in \mathbb{R}^n$ on $G$ to signals $x_c \in \mathbb{R}^{n_c}$ on $G_c$. For this we associate a projection matrix $\boldsymbol{P} \in \mathbb{R}^{n_c \times n}$ and it's corresponding Moore-Penrose pseudoinverse $\boldsymbol{P}^+ \in \mathbb{R}^{n \times n_c}$ with $\mathcal{P}$, defined as

$$\boldsymbol{P}[p, i] = \begin{cases} \frac{1}{|\mathcal{V}^{(p)}|} & \text{if } v^{(i)} \in \mathcal{V}^{(p)} \\ 0 & \text{otherwise} \end{cases}, \qquad \boldsymbol{P}^+[i, p] = \begin{cases} 1 & \text{if } v^{(i)} \in \mathcal{V}^{(p)} \\ 0 & \text{otherwise} \end{cases}.$$

For a graph Laplacian $\boldsymbol{L}$ and this choice of $\boldsymbol{P}$, Loukas (2018) shows that $\boldsymbol{L}_c = \boldsymbol{P}^\mp \boldsymbol{L} \boldsymbol{P}^+$ [3] is the Laplacian of the coarsened graph $G_c$ with weight matrix

$$\boldsymbol{W}_c[p, q] = \sum_{v^{(i)} \in \mathcal{V}^{(p)}, v^{(j)} \in \mathcal{V}^{(q)}} \boldsymbol{W}[i, j]. \tag{6}$$

To measure the similarity between the Laplacians $\boldsymbol{L}$ and $\boldsymbol{L}_c$, Loukas (2018) introduces the following fairly general notion of a restricted spectral approximation.

**Definition 5 (Restricted Spectral Approximation)** *Two matrices $\boldsymbol{L} \in \mathbb{R}^{n \times n}$ and $\boldsymbol{L}_c = \boldsymbol{P}^\mp \boldsymbol{L} \boldsymbol{P}^+ \in \mathbb{R}^{n_c \times n_c}$ are $(\mathbb{V}, \epsilon)$-similar, with $\mathbb{V}$ being a $k \leq n_c \leq n$-dimensional subspace of $\mathbb{R}^n$, if there exists an $\epsilon \geq 0$ such that*

$$\left\| \boldsymbol{x} - \boldsymbol{P}^+ \boldsymbol{P} \boldsymbol{x} \right\|_{\boldsymbol{L}} \leq \epsilon \left\| \boldsymbol{x} \right\|_{\boldsymbol{L}} \quad \forall \boldsymbol{x} \in \mathbb{V},$$

*where $\|\boldsymbol{x}\|_{\boldsymbol{L}} = \sqrt{\boldsymbol{x}^T \boldsymbol{L} \boldsymbol{x}}$ denotes a $\boldsymbol{L}$-induced semi-norm.*

When $\mathbb{V}$ is chosen as the span of the $k$ eigenvectors associated with the $k$ smallest non-zero eigenvalues of $\boldsymbol{L}$, $\mathbb{U}_k = \text{span}(\boldsymbol{u}_1, \dots, \boldsymbol{u}_k) = \text{span}(\boldsymbol{U}_k)$, then if $\boldsymbol{L}$ and $\boldsymbol{L}_c$ are $(\mathbb{U}_k, \epsilon)$-similar, Loukas (2018) shows that the $k$ smallest non-zero eigenvalues of $\boldsymbol{L}_c$ are close to the ones of $\boldsymbol{L}$, and the lifted eigenvectors $\boldsymbol{P}^\top \boldsymbol{u}_i$ are aligned with the original eigenvectors $\boldsymbol{u}_i$.

### C.1  LOCAL VARIATION COARSENING

Loukas (2018) further proposes a multi-level graph coarsening algorithm to construct a coarsening sequence $G = G_0, G_1, \dots$, with each coarse graph $G_l$ possessing a Laplacian $\boldsymbol{L}_l$ with high spectral similarity to the original Laplacian $\boldsymbol{L}$ associated with $G$, according to Definition 5. While the procedure in the original work is quite general and allows for any choice of $\mathbb{V}$, let us commence by considering the case where $\mathbb{V}$ is chosen as the span of the $k$ eigenvectors associated with the $k$ smallest non-zero eigenvalues of $\boldsymbol{L}$. In this case, for each coarsening step, the algorithm first computes a local variation cost for each contraction set and then contracts disjoint sets with small cost. The local variation cost of a contraction set $\mathcal{C} \subseteq \mathcal{V}$ is defined as

$$c_{\text{lv}}(G, \mathcal{C}) = \frac{\left\| \Pi_{\mathcal{C}}^\perp \boldsymbol{A}_{l-1} \right\|_{\boldsymbol{L}_\mathcal{C}}^2}{|\mathcal{C}| - 1}. \tag{7}$$

---

[3] $\boldsymbol{P}^\mp$ denotes the transpose of the Moore-Penrose pseudoinverse of $\boldsymbol{P}$.

Here, $\Pi_{\mathcal{C}}^{\perp} = \boldsymbol{I} - \boldsymbol{P}_{\mathcal{C}}^{+}\boldsymbol{P}_{\mathcal{C}}$ is the complementary projection matrix when contracting solely the nodes in $\mathcal{C}$. $\boldsymbol{L}_{\mathcal{C}}$ is the Laplacian of $G_{l-1}$ with modified weights $\boldsymbol{W}_{\mathcal{C}}$, given by

$$\boldsymbol{W}_{\mathcal{C}}[i,j] = \begin{cases} \boldsymbol{W}[i,j] & \text{if } v^{(i)}, v^{(j)} \in \mathcal{C} \\ 2 \cdot \boldsymbol{W}[i,j] & \text{if } v^{(i)} \in \mathcal{C}, v^{(j)} \notin \mathcal{C} \\ 0 & \text{otherwise.} \end{cases}$$

$\boldsymbol{A}_{l-1}$ is a matrix allowing us to evaluate the cost only on the selected Laplacian subspace $\mathbb{U}_k$. It is defined recursively as

$$\boldsymbol{A}_0 = \boldsymbol{U}_k \boldsymbol{\Lambda}_k^{+1/2}, \qquad \boldsymbol{A}_l = \boldsymbol{B}_l (\boldsymbol{B}_l^{\top} \boldsymbol{L}_l \boldsymbol{B}_l)^{+1/2},$$

with $\boldsymbol{B}_l = \boldsymbol{P}_l \boldsymbol{P}_{l-1} \dots \boldsymbol{P}_1 \boldsymbol{A}_0$ being the first-level matrix $\boldsymbol{A}_0$ left-multiplied by the projection matrices $\boldsymbol{P}_i$ of all coarsening steps up to level $l$.

With this construction, it holds that the Laplacian of the resulting coarsened graph $G_l$ is $(\mathbb{U}_k, \epsilon)$-similar to the Laplacian of $G$ with $\epsilon \leq \prod_{i=1}^{l}(1 + \sigma_i) - 1$, where $\sigma_i$ bounds the error introduced by the $i$-th coarsening step as

$$\sigma_i^2 \leq \sum_{\mathcal{C} \in \mathcal{P}_i} \left\| \Pi_{\mathcal{C}}^{\perp} \boldsymbol{A}_{l-1} \right\|_{\boldsymbol{L}_{\mathcal{C}}}^2,$$

with $\mathcal{P}_i$ being the node partitioning used in the $i$-th coarsening step.

## C.2 Adapting Local Variation Coarsening to our Setting

A noteworthy distinction between our setting and the one investigated by Loukas (2018) lies in the usage of unweighted graphs. Therefore, we assign a weight of 1 to each edge in the original graph $G$. For a coarsening sequence $G = G_0, G_1, \dots$, we then maintain a sequence of weight matrices according to Equation 6. With this, all the quantities required to calculate the local variation cost are readily available and it can be employed to instantiate the cost function $c$ in Algorithm 1.

An important point to mention is that while the method proposed by Loukas (2018) operates deterministically, our approach introduces an element of randomness.

## D   COARSENING SEQUENCE SAMPLING

This section describes the methodology used to sample a coarsening sequence $\pi \in \Pi_{\mathcal{F}}(G)$ for a given graph $G$. Importantly, this implicitly defines the coarsening sequence probability density function $q(\pi \mid G)$.

Our approach for sampling a coarsening sequence $\pi \in \Pi_{\mathcal{F}}(G)$ from $q(\pi \mid G)$ is presented in Algorithm 1. At each coarsening step $l$, we assess the cost of all contraction sets $\mathcal{F}(G_l)$, where a lower cost indicates a preference for contraction. This cost computation is based on the current graph $G_l$ and may incorporate information from previous coarsening steps. While there are no inherent restrictions on the cost function, it should be efficiently computable for practical purposes. Subsequently, we randomly sample a reduction fraction. Then, employing a greedy and randomized strategy, we select a cost-minimizing partition of $\mathcal{F}(G_l)$ that achieves the desired reduction fraction upon contracting the graph as per Definition 3. This partitioning procedure is detailed in Algorithm 2. The resulting contraction sets are used to construct the coarsened graph $G_{l-1}$ in accordance with Definition 3.

Note that our procedures are designed with a heuristic approach that allows alternative choices. Various components of the procedure remain parametric, such as the contraction family $\mathcal{F}$, the cost function $c$, the range of reduction fractions $[\rho_{\min}, \rho_{\max}]$, and the randomization parameter $\lambda$.

**Practical Considerations**   Depending on the structure of the graph, it may not always be feasible to achieve a non-overlapping partitioning of the node set that satisfies the desired reduction fraction. Our empirical investigations indicate that such circumstances rarely arise in the examined datasets, provided that $\rho_{\max}$ is not unreasonably large. In the event that such a situation does occur, we choose to proceed with the partitioning achieved thus far. Furthermore, to mitigate an imbalance of numerous small graphs in the coarsening sequence, we deterministically set the reduction fraction $\rho$ to $\rho_{\max}$ when the current graph contains fewer than 16 nodes. It should be noted that during the training phase, we sample a coarsening sequence from the dataset graph but only consider the graph at a random level within the sequence. Consequently, in terms of practical implementation, Algorithm 1 does not return the coarsening sequence $\pi$, but instead provides a coarsened graph along with the node and edge features necessary for model training. To optimize computational efficiency, upon computation of the coarsening sequence, we cache its elements, select a level at random, and return the corresponding graph and features, subsequently removing this element from the cache. Recomputation of the coarsening sequence for a specific graph is necessitated only when the cache is exhausted.

**Hyperparameters**   In all the experiments conducted described in Section 4, we use the *Local Variation Cost* 7 (presented in Appendix C) with a preserving eigenspace size of $k = 8$ as the cost function $c$. The range of reduction fractions is set as $[\rho_{\min}, \rho_{\max}] = [0.1, 0.3]$. The randomization parameter $\lambda$ is assigned a value of $0.3$. Moreover, edge contractions are used for all graphs, i.e., $\mathcal{F}(G) = \mathcal{E}$, for a graph $G = (\mathcal{V}, \mathcal{E})$.

### D.1   COST FUNCTION ABLATION STUDY

We assess the influence of the cost function $c$ on the generative performance of our model. In this regard, we compare the *Local Variation Cost* 7 with its specified parameters against a random cost function, where each contraction set is assigned a uniformly sampled random cost from the range $[0, 1]$. The randomization parameter $\lambda$ is set to $0$ for the random cost function. Otherwise, the prescribed parameters are used for both cost functions. Following this comparison, we train our model on the planar, tree, and protein datasets described in Section 4. We report the average ratio between the generated and training set MMD values. For the synthetic datasets, we additionally report the fraction of valid, unique, and novel graphs among the generated graphs. The results of the experiments are shown in Table 3. A key observation from the results is that the *Local Variation Cost* demonstrates a slightly superior performance over the random cost function across all evaluated datasets. This suggests that while spectrum-preserving coarsening contributes positively to the model's performance, it is not an indispensable factor and alternative cost functions could potentially be suitable.

**Algorithm 1 Random Coarsening Sequence Sampling:** This algorithm demonstrates the process of Random Coarsening Sequence Sampling, detailing how a coarsening sequence is sampled for a given graph. Starting with the initial graph, it iteratively computes the costs of all possible contraction sets, samples a reduction fraction, and uses a greedy randomized strategy to find a cost-minimizing partition of the contraction sets. Then a coarsening of the preceding graph is added to the coarsening sequence, according to Definition 3, and the process repeats until the graph is reduced to a single node.

---

**Parameters:** contraction family $\mathcal{F}$, cost function $c$, reduction fraction range $[\rho_{\min}, \rho_{\max}]$
**Input:** graph $G$
**Output:** coarsening sequence $\pi = (G_0, \ldots, G_L) \in \Pi_{\mathcal{F}}(G)$
1: **function** RNDREDSEQ($G$)
2: $\quad$ $G_0 = (\mathcal{V}_0, \mathcal{E}_0) \leftarrow G$
3: $\quad$ $\pi \leftarrow (G_0)$
4: $\quad$ $l \leftarrow 0$
5: $\quad$ **while** $|\mathcal{V}_{l-1}| > 1$ **do**
6: $\quad\quad$ $l \leftarrow l + 1$
7: $\quad\quad$ $\rho \sim \text{Uniform}([\rho_{\min}, \rho_{\max}])$ $\qquad\qquad\qquad\qquad$ ▷ random reduction fraction
8: $\quad\quad$ $f \leftarrow c(\cdot, G_0, (\mathcal{P}_1, \ldots, \mathcal{P}_{l-1}))$ $\qquad$ ▷ cost function depending on previous contractions
9: $\quad\quad$ $m \leftarrow \lceil \rho \cdot |\mathcal{V}_{l-1}| \rceil$ $\qquad\qquad\qquad\qquad\qquad$ ▷ reduction amount
10: $\quad\quad$ $\mathcal{P}_l \leftarrow \text{RNDGREEDYMINCOSTPART}(\mathcal{F}(G_{l-1}), f, m)$
11: $\quad\quad$ $\mathcal{V}_l \leftarrow \{v_l^{(p)} \mid \mathcal{V}_{l-1}^{(p)} \in \mathcal{P}_l\}$ $\qquad\qquad\qquad\qquad\qquad$ ▷ new node set
12: $\quad\quad$ $\mathcal{E}_l \leftarrow \{e_l^{\{p,q\}} \mid e_{l-1}^{\{i,j\}} \in \mathcal{E}_{l-1}, v_{l-1}^{(i)} \in \mathcal{V}_l^{(p)}, v_{l-1}^{(j)} \in \mathcal{V}_l^{(q)}\}$ $\quad$ ▷ new edge set
13: $\quad\quad$ $G_l \leftarrow (\mathcal{V}_l, \mathcal{E}_l)$
14: $\quad$ **end while**
15: $\quad$ **return** $(G_0, \ldots, G_l)$ $\qquad\qquad\qquad\qquad\qquad$ ▷ Return coarsening sequence
16: **end function**

---

**Algorithm 2 Randomized Greedy Min-Cost Partitioning:** This algorithm represents a Randomized Greedy Min-Cost Partitioning, where the cheapest contraction set is selected iteratively from the candidate sets. The selected set, with probability $1 - \lambda$, is discarded and the process continues with the next cheapest set. Conversely, with probability $\lambda$, the set is added to the partitioning and all intersecting sets are removed from the remaining candidates. Termination occurs when the partitioning contains at least $m$ sets or there are no remaining candidates.

---

**Parameters:** randomization parameter $\lambda \in [0, 1]$
**Input:** candidate contraction sets $\mathcal{C} = \{\mathcal{V}^{(1)}, \ldots, \mathcal{V}^{(n)}\}$, reduction amount $m$
**Output:** partitioning $\mathcal{P} = \{\mathcal{V}^{(1)}, \ldots, \mathcal{V}^{(m)}\} \subseteq \mathcal{C}$, with $\mathcal{V}^{(i)} \cap \mathcal{V}^{(j)} = \emptyset$ for $i \neq j$ and $\sum_{\mathcal{V} \in \mathcal{P}} |\mathcal{V}| - |\mathcal{P}| \geq m$
1: **function** RNDGREEDYMINCOSTPART($\mathcal{C}, f, m$)
2: $\quad$ $\mathcal{P} \leftarrow \emptyset$
3: $\quad$ **while** $\sum_{\mathcal{V} \in \mathcal{P}} |\mathcal{V}| - |\mathcal{P}| < m$ and $\mathcal{C} \neq \emptyset$ **do**
4: $\quad\quad$ **repeat**
5: $\quad\quad\quad$ $\mathcal{V}^* \leftarrow \arg\min_{\mathcal{V} \in \mathcal{C}} f(\mathcal{V})$
6: $\quad\quad\quad$ $\mathcal{C} \leftarrow \mathcal{C} \setminus \{\mathcal{V}^*\}$
7: $\quad\quad\quad$ $b \sim \text{Bernoulli}(\lambda)$
8: $\quad\quad$ **until** $b = 0$ or $\mathcal{C} = \emptyset$
9: $\quad\quad$ $\mathcal{P} \leftarrow \mathcal{P} \cup \{\mathcal{V}^*\}$
10: $\quad\quad$ $\mathcal{C} \leftarrow \{\mathcal{V} \in \mathcal{C} \mid \mathcal{V} \cap \mathcal{V}^* = \emptyset\}$
11: $\quad$ **end while**
12: $\quad$ **return** $\mathcal{P}$
13: **end function**

| | Planar graphs | | Tree graphs | | Proteins |
|---|---|---|---|---|---|
| Cost | Ratio ↓ | V.U.N. ↑ | Ratio ↓ | V.U.N. ↑ | Ratio ↓ |
| Local Variation | 2.1 | 95.0 | 4.0 | 100 | 5.9 |
| Random | 3.9 | 85.0 | 6.2 | 100 | 8.2 |

Table 3: Ablation study of the cost function $c$.

# E    DENOISING DIFFUSION

Denoising diffusion models represent a class of generative models initially proposed for image generation (Sohl-Dickstein et al., 2015; Ho et al., 2020) and subsequently adapted for various data modalities, such as audio (Du et al., 2023; Borsos et al., 2023; Agostinelli et al., 2023), text (Austin et al., 2021; Hoogeboom et al., 2021), discretized images (Gu et al., 2022; Tang et al., 2022; Augustin et al., 2022; Hoogeboom et al., 2021), and graphs (Vignac et al., 2023a; Haefeli et al., 2023), exhibiting exceptional performance across domains. These models are founded on the fundamental concept of learning to invert an iterative stochastic data degradation process. Consequently, denoising diffusion models comprise two main components: (1) a predefined fixed forward process, also known as the noising process $\{x_t\}_{t=0}^T$, which progressively transforms samples drawn from the data generating distribution $x_0 = x \sim p(x)$ into noisy samples $x_t \sim p(x_t)$, with the level of degradation noise increasing as $t$ progresses; and (2) a trainable backward process, referred to as the denoising process, which aims to reverse the effects of the noising process. The construction of the forward process ensures that the limit distribution $p(x_T)$ is both known and simple, for example, a Gaussian, thereby allowing for easy sampling. During inference, pure noise samples are drawn from this distribution and subsequently transformed by the denoising process to obtain samples following the data-generating distribution $p(x)$.

## E.1    GENERATIVE MODELING WITH STOCHASTIC DIFFERENTIAL EQUATIONS

Song et al. (2021) introduce a continuous-time formulation of the diffusion model, in which forward and backward processes are described by stochastic differential equations. In this summary, we outline the general framework along with specific instantiations of its components and improvements proposed by Karras et al. (2022). The same framework setup is adopted in the work of Yan et al. (2023).

The fundamental concept involves modeling a continuous-time indexed diffusion process $\{\boldsymbol{x}_t\}_{t=0}^T$ as the solution to an Itô stochastic differential equation of the form

$$\mathrm{d}\boldsymbol{x}_t = \boldsymbol{f}(\boldsymbol{x}_t, t)\mathrm{d}t + g(t)\mathrm{d}\boldsymbol{w}, \tag{8}$$

where $\boldsymbol{f}$, and $g$ are the drift and diffusion coefficients, respectively, and $\boldsymbol{w}$ is a standard Wiener process. Different choices of $\boldsymbol{f}$ and $g$ lead to distinct diffusion processes with varying dynamics. We adopt the *variance exploding* setting, where $\boldsymbol{f}(\boldsymbol{x}_t, t) = \boldsymbol{0}$ and $g(t) = \frac{\mathrm{d}\sigma^2(t)}{\mathrm{d}t}$, for a time-dependent noise scale $\sigma(t)$. Consistent with Karras et al. (2022), we select $\sigma(t)$ to be linear in $t$. This leads to the following forward process:

$$\mathrm{d}\boldsymbol{x}_t = \sqrt{2t}\mathrm{d}\boldsymbol{w}. \tag{9}$$

When integrating from time $0$ to $t$, the corresponding transition distribution is given by:

$$p(\boldsymbol{x}_t \mid \boldsymbol{x}_0) = N(\boldsymbol{x}_t; \boldsymbol{x}_0, t^2\boldsymbol{I}). \tag{10}$$

The stochastic trajectory of a sample behaving according to 8, but with reversed time direction, can be described by a stochastic differential equation too (Anderson, 1982):

$$\mathrm{d}\boldsymbol{x}_t = [\boldsymbol{f}(\boldsymbol{x}_t, t) - g(t)^2 \nabla_{\boldsymbol{x}_t} \log p(\boldsymbol{x}_t)]\mathrm{d}t + g(t)\mathrm{d}\bar{\boldsymbol{w}}, \tag{11}$$

With the prescribed instantiations of $\boldsymbol{f}$ and $g$, the backward process becomes

$$\mathrm{d}\boldsymbol{x}_t = -2t\nabla_{\boldsymbol{x}_t} \log p(\boldsymbol{x}_t)\mathrm{d}t + \sqrt{2t}\mathrm{d}\bar{\boldsymbol{w}}. \tag{12}$$

Here, $\bar{\boldsymbol{w}}$ denotes a Wiener process with reversed time direction, i.e., from $T$ to $0$. We get a generative model, by simulating this process from time $T$ to $0$, to transform a sample from the prior distribution $p(\boldsymbol{x}_T)$ into one that follows the data distribution $p(\boldsymbol{x}_0)$.

**Denoising Score Matching**    Simulating the backward process necessitates evaluating the gradient of the log-density $\nabla_{\boldsymbol{x}_t} \log p(\boldsymbol{x}_t)$. As this gradient is generally intractable, we train a model to approximate it using denoising score matching. This training principle is based on the insight that conditioned on $\boldsymbol{x}_0$, the gradient of $\log p(\boldsymbol{x}_t \mid \boldsymbol{x}_0)$ with respect to $\boldsymbol{x}_t$ can be expressed as:

$$\nabla_{\boldsymbol{x}_t} \log p(\boldsymbol{x}_t \mid \boldsymbol{x}_0) = \frac{\boldsymbol{x}_0 - \boldsymbol{x}_t}{t^2}. \tag{13}$$

Vincent (2011) proof that for fixed $t$ the minimizer $S_{\theta^\star}(\boldsymbol{x})$ of

$$\mathbb{E}_{p(\boldsymbol{x}_0)} \mathbb{E}_{p(\boldsymbol{x}_t|\boldsymbol{x}_0)} \left[ \left\| S_{\boldsymbol{\theta}}(\boldsymbol{x}_t) - \frac{\boldsymbol{x}_0 - \boldsymbol{x}_t}{t^2} \right\|_F^2 \right] \tag{14}$$

satisfies $S_{\boldsymbol{\theta}^\star}(\boldsymbol{x}) = \nabla_{\boldsymbol{x}_t} \log p(\boldsymbol{x}_t)$ almost surely.

Instead of approximating the score directly, Karras et al. (2022) train a denoising model $D_{\boldsymbol{\theta}}(\boldsymbol{x}_t, t)$ to reconstruct $\boldsymbol{x}_0$ from $\boldsymbol{x}_t$. The joint training objective for all $t$ is given by:

$$\mathbb{E}_t \left[ \lambda(t) \mathbb{E}_{p(\boldsymbol{x}_0)} \mathbb{E}_{p(\boldsymbol{x}_t|\boldsymbol{x}_0)} \left[ \|D_{\boldsymbol{\theta}}(\boldsymbol{x}_t, t) - \boldsymbol{x}_0\|_F^2 \right] \right]. \tag{15}$$

Here, $\lambda : [0, T] \to \mathbb{R}_+$ is a time-dependent weighting function, and $t$ is sampled such that $\ln(t) \sim N(P_{\text{mean}}, P_{\text{std}})$ (details in Table 4). The score can then be recovered from the optimal denoising model $D_{\boldsymbol{\theta}}^*$ as $\nabla_{\boldsymbol{x}_t} \log p(\boldsymbol{x}_t) = (D_{\boldsymbol{\theta}}^*(\boldsymbol{x}_t, t) - \boldsymbol{x}_t)/t^2$.

**Preconditioning**   Instead of representing $D_{\boldsymbol{\theta}}$ directly as a neural network, Karras et al. (2022) propose preconditioning a network $F_{\boldsymbol{\theta}}$ with a time-dependent skip connection to improve the training dynamics. Furthermore, they aim to achieve uniformity in the variance of the input and output of the network by employing appropriate scaling:

$$D_{\boldsymbol{\theta}}(\boldsymbol{x}_t, t) = c_{\text{skip}}(t)\boldsymbol{x}_t + c_{\text{out}}(t)F_{\boldsymbol{\theta}}(c_{\text{in}}(t)\boldsymbol{x}_t, t).$$

Weighting functions are summarized in Table 4.

**Self-conditioning**   Sampling from the diffusion model is an iterative procedure, in which the denoising model is iteratively queried to construct samples $\boldsymbol{x}_{t'}$ from $\boldsymbol{x}_t$, for $t' < t$. Chen et al. (2023a) propose to additionally condition $D_{\boldsymbol{\theta}}$ on previous estimates $\hat{\boldsymbol{x}}_{t'}$ for improved sample quality. We observed that this improves the generative performance of the model and thus adopt this technique in our work. Consequently, we augment the denoising model with an additional input as

$$D_{\boldsymbol{\theta}}(\boldsymbol{x}_t, \hat{\boldsymbol{x}}, t) = c_{\text{skip}}(t)\boldsymbol{x}_t + c_{\text{out}}(t)F_{\boldsymbol{\theta}}(c_{\text{in}}(t)\boldsymbol{x}_t, c_{\text{self}}(t)\hat{\boldsymbol{x}}, t).$$

During training, given a noisy sample $\boldsymbol{x}_t$, we then set $\hat{\boldsymbol{x}} = \boldsymbol{0}$ with 50% probability, and $\hat{\boldsymbol{x}} = D_{\boldsymbol{\theta}}(\boldsymbol{x}_t, \boldsymbol{0}, t)$ otherwise. Note that in the latter case, gradients are not propagated through $\hat{\boldsymbol{x}}$. During sampling, we set $\hat{\boldsymbol{x}}$ to a previous estimate.

**Sampling**   Once the denoising model is trained, we can compute the gradient of the log-density and simulate the backward process, starting from a sample from the prior distribution $p(\boldsymbol{x}_T)$. Stochastic sampling based on Heun's 2nd order method was proposed by Karras et al. (2022). The procedure, combined with self-conditioning, is summarized in Algorithm 3. Notably, this algorithm is identical to the one presented in Yan et al. (2023).

**Parametrizing Graph Expansion**   We use this framework to model the conditional distribution $p_{\boldsymbol{\theta}}(\boldsymbol{x}_l \mid \boldsymbol{x}_{l-1})$ over the node and edge features required to refine and subsequently expand the graph $\tilde{G}_l$. For this, continuous representations of the node and edge features are required. For our experiments, where both node and edge features are binary, we represent 0 values as $-1$ and 1 values as 1. Let $(\boldsymbol{v}_l)_0$ and $(\boldsymbol{e}_l)_0$ denote the corresponding continuous representations of the node and edge features. We instantiate the diffusion framework jointly over these features. It should be noted that as the diffusion process adds noise independently to each dimension, this is equivalent to organizing the node and edge features in a single vector $\boldsymbol{x} = [\boldsymbol{v}, \boldsymbol{e}] \in [-1, 1]^{\tilde{n}_l + \tilde{m}_l}$ that is then used in the diffusion framework.

We further instantiate the denoising model $D_{\boldsymbol{\theta}}$, specifically its preconditioned version, with a graph neural network $\text{GNN}_{\boldsymbol{\theta}}$ (such as our proposed *Local PPGN* model) that operates on the graph $\tilde{G}_l$. This can be expressed as:

$$F_{\boldsymbol{\theta}}(\boldsymbol{x}_t, \hat{\boldsymbol{x}}, t) = \text{GNN}_{\boldsymbol{\theta}}([(\boldsymbol{v}_l)_t, (\boldsymbol{e}_l)_t], [\hat{\boldsymbol{v}}_l, \hat{\boldsymbol{e}}_l], t, \tilde{G}_l).$$

Here, $(\boldsymbol{v}_l)_t$ and $(\boldsymbol{e}_l)_t$ denote noisy samples of the node and edge features, respectively, and $\hat{\boldsymbol{x}} = [\hat{\boldsymbol{v}}_l, \hat{\boldsymbol{e}}_l]$ represents a previous estimate of the node and edge features that is concatenated to the input of the GNN.

---

**Algorithm 3 SDE Sampling:** This describes the sampling procedure by simulating the backward process using a given denoising model.

---

**Parameters:** number of steps $N$, time schedule $\{t_i\}_{i=0}^{N}$, noise addition schedule $\{\gamma_i\}_{i=1}^{N-1}$
**Input:** denoising model $D_{\boldsymbol{\theta}}$
**Output:** sample $\boldsymbol{x} = [\boldsymbol{v}_l, \boldsymbol{e}_l]$

1: **function** SDESAMPLE($D_{\boldsymbol{\theta}}$)
2:    $\boldsymbol{x} \sim N(\boldsymbol{0}, \sigma_{\max}^2 \boldsymbol{I})$               $\triangleright$ Sample from prior
3:    $\hat{\boldsymbol{x}} \leftarrow \boldsymbol{0}$
4:    **for** $i = 1, \ldots, N$ **do**
5:      $\boldsymbol{\epsilon} \sim N(\boldsymbol{0}, S_{\text{noise}}^2 \boldsymbol{I})$
6:      $\tilde{t} \leftarrow t_i + \gamma_i t_i$
7:      $\tilde{\boldsymbol{x}} \leftarrow \boldsymbol{x} + \sqrt{\tilde{t}_i^2 - t_i^2}\,\boldsymbol{\epsilon}$           $\triangleright$ Add noise
8:      $\hat{\boldsymbol{x}} \leftarrow D_{\boldsymbol{\theta}}(\tilde{\boldsymbol{x}}, \hat{\boldsymbol{x}}, \tilde{t})$           $\triangleright$ Denoise
9:      $\boldsymbol{d} \leftarrow (\tilde{\boldsymbol{x}} - \hat{\boldsymbol{x}})/\tilde{t}$         $\triangleright$ Compute gradient
10:      $\boldsymbol{x} \leftarrow \tilde{\boldsymbol{x}} + (t_{i+1} - \tilde{t})\boldsymbol{d}$     $\triangleright$ Euler step from $\tilde{t}$ to $t_{i+1}$
11:      **if** $t_{i+1} > 0$ **then**
12:        $\hat{\boldsymbol{x}} \leftarrow D_{\boldsymbol{\theta}}(\boldsymbol{x}, \hat{\boldsymbol{x}}, t_{i+1})$       $\triangleright$ Denoise
13:        $\boldsymbol{d}' \leftarrow (\boldsymbol{x} - \hat{\boldsymbol{x}})/t_{i+1}$      $\triangleright$ Compute gradient
14:        $\boldsymbol{x} \leftarrow \tilde{\boldsymbol{x}} + (t_{i+1} - \tilde{t}) \cdot \frac{1}{2}(\boldsymbol{d} + \boldsymbol{d}')$    $\triangleright$ Second order correction
15:      **end if**
16:    **end for**
17:    **return** $\boldsymbol{x}$
18: **end function**

---

For a given expansion $\tilde{G}_l$ of a reduced graph with target node and edge encodings $\boldsymbol{v}_l$ and $\boldsymbol{e}_l$, during training, we sample a time-step $t$, construct a noisy sample $\boldsymbol{x}_t = \boldsymbol{x}_0 + t \cdot \boldsymbol{\epsilon}$, with $\boldsymbol{\epsilon} \sim N(\boldsymbol{0}, \boldsymbol{I})$, and optimize the model parameters by minimizing the l2 loss between the prediction and the target $\boldsymbol{x}_0 = [\boldsymbol{v}_l, \boldsymbol{e}_l]$. Overall, training involves sampling a coarsening sequence out of which a random level $l$ is selected. Subsequently, a noisy sample is constructed and the model is trained to denoise it. The objective in 15 is referred to as *denoising score matching*. It is known to be equivalent to a reweighed variational bound (Ho et al., 2020). As such, our training procedure can be interpreted as maximizing a weighted lower bound of random terms in Equation 3. Algorithm 4 summarizes the loss computation for given node and edge features $\boldsymbol{v}_l$ and $\boldsymbol{e}_l$ and denoising model $D_{\boldsymbol{\theta}}$ that is assumed to be instantiated with a graph neural network $\text{GNN}_{\boldsymbol{\theta}}$ operating on underlying graph $\tilde{G}_l$.

| | | |
|---|---|---|
| Constants | $\sigma_{\text{data}} = 0.5$ 
 $\sigma_{\min} = 0.002$ 
 $P_{\text{mean}} = -1.2$ 
 $S_{\text{tmin}} = 0.05$ 
 $S_{\text{noise}} = 1.003$ | $\rho = 7$ 
 $\sigma_{\max} = 80$ 
 $P_{\text{std}} = 1.2$ 
 $S_{\text{tmax}} 50$ 
 $S_{\text{churn}} = 40$ |
| Weightings | $c_{\text{in}}(t) = \frac{1}{\sigma_{\text{data}}^2 + t^2}$ 

 $c_{\text{skip}}(t) = \frac{\sigma_{\text{data}}^2}{\sigma_{\text{data}}^2 + t^2}$ | $c_{\text{out}}(t) = \frac{t \cdot \sigma_{\text{data}}}{\sqrt{\sigma_{\text{data}}^2 + t^2}}$ 

 $c_{\text{self}}(t) = \sigma_{\text{data}}$ |
| Schedules | $t_i = (\sigma_{\max}^{\frac{1}{\rho}} + \frac{i}{N-1}(\sigma_{\min}^{\frac{1}{\rho}} - \sigma_{\max}^{\frac{1}{\rho}}))^{\rho}$ | $\gamma_i = \mathbf{1}_{S_{\text{tmin}} \leq t_i \leq S_{\text{tmax}}} \cdot \min(\frac{S_{\text{churn}}}{N}, \sqrt{2} - 1)$ |

Table 4: Summary of hyperparameters for the diffusion process.

---

**Algorithm 4 Diffusion loss:** This describes the loss computation for given node and edge features $\boldsymbol{v}_l$ and $\boldsymbol{e}_l$.

---

**Input:** node and edge features $\boldsymbol{v}_l$ and $\boldsymbol{e}_l$, denoising model $D_{\boldsymbol{\theta}}$
**Output:** trained model parameters $\boldsymbol{\theta}$

1: **function** DIFFUSIONLOSS($\boldsymbol{v}_l, \boldsymbol{e}_l, D_{\boldsymbol{\theta}}$)
2:     $(\boldsymbol{v}_l)_0 \leftarrow$ ENCODE($\boldsymbol{v}_l$)                               ▷ Encode node features
3:     $(\boldsymbol{e}_l)_0 \leftarrow$ ENCODE($\boldsymbol{e}_l$)                               ▷ Encode edge features
4:     $\ln(t) \sim N(P_{\text{mean}}, P_{\text{std}})$                               ▷ Sample noise level
5:     **if** $b = 1$, where $b \sim$ Bernoulli$(0.5)$ **then**
6:         $(\boldsymbol{e}_l)_t \leftarrow (\boldsymbol{e}_l)_0 + t \cdot \boldsymbol{\epsilon}$, with $\boldsymbol{\epsilon} \sim N(\boldsymbol{0}, \boldsymbol{I})$
7:         $(\boldsymbol{v}_l)_t \leftarrow (\boldsymbol{v}_l)_0 + t \cdot \boldsymbol{\epsilon}$, with $\boldsymbol{\epsilon} \sim N(\boldsymbol{0}, \boldsymbol{I})$
8:         $\hat{\boldsymbol{v}}, \hat{\boldsymbol{e}} \leftarrow D_{\boldsymbol{\theta}}([(\boldsymbol{v}_l)_t, (\boldsymbol{e}_l)_t], [\boldsymbol{0}, \boldsymbol{0}], t)$                  ▷ Get self-conditioning estimates
9:         clip gradients of $\hat{\boldsymbol{v}}$ and $\hat{\boldsymbol{e}}$
10:     **else**
11:         $\hat{\boldsymbol{v}}, \hat{\boldsymbol{e}} \leftarrow \boldsymbol{0}, \boldsymbol{0}$
12:     **end if**
13:     $(\boldsymbol{e}_l)_t \leftarrow (\boldsymbol{e}_l)_0 + t \cdot \boldsymbol{\epsilon}$, with $\boldsymbol{\epsilon} \sim N(\boldsymbol{0}, \boldsymbol{I})$
14:     $(\boldsymbol{v}_l)_t \leftarrow (\boldsymbol{v}_l)_0 + t \cdot \boldsymbol{\epsilon}$, with $\boldsymbol{\epsilon} \sim N(\boldsymbol{0}, \boldsymbol{I})$
15:     $\hat{\boldsymbol{v}}, \hat{\boldsymbol{e}} \leftarrow D_{\boldsymbol{\theta}}([(\boldsymbol{v}_l)_t, (\boldsymbol{e}_l)_t], [\hat{\boldsymbol{v}}, \hat{\boldsymbol{e}}], t)$                               ▷ Denoise
16:     **return** $\|\hat{\boldsymbol{v}} - (\boldsymbol{v}_l)_0\|_F^2 + \|\hat{\boldsymbol{e}} - (\boldsymbol{e}_l)_0\|_F^2$
17: **end function**

---

## F  GRAPH NEURAL NETWORKS

A key element of our generative methodology is the neural architecture, employed to parameterize the denoising model. This architecture necessitates numerous desirable characteristics: it should be expressive enough to adequately model the complex joint distribution of node and edge features, maintain invariance to the permutation of nodes, and sustain a low computational expense. The importance of this final attribute is underscored by our iterative expansion strategy, which frees us from the need to model a distribution over all conceivable $\mathcal{O}(n^2)$ node pairings. As a result, our objective is to identify a model that scales at a rate less than quadratic with respect to the total number of nodes.

Despite the rich literature on graph neural networks, addressing all these requirements simultaneously presents a considerable challenge. Message passing GNNs, although linear in complexity with respect to the number of edges, are shown to be no more expressive than the 1-Weisfeiler-Lehman test (Xu et al., 2019). Additionally, handling fully connected graphs effectively is a hurdle for them that might impede the processing of locally dense expanded graphs (Definition 1). In contrast, higher-order GNNs, while being more expressive, come with elevated computational costs. Our investigation featured various architectures, including message passing GNNs, a transformer-based model advocated by Vignac et al. (2023a), and *PPGN* (Maron et al., 2020). Our experimentation revealed that *PPGN* performed exceptionally well, although being computationally expensive.

### F.1  PROVABLY POWERFUL GRAPH NETWORKS

Maron et al. (2020) introduce *PPGN*, a permutation-equivariante graph neural network architecture. The *PPGN* possesses provable 3-WL expressivity, which is categorically stronger than message passing models and operates on dense graph representations.

Given a graph embedding $\mathbf{H} \in \mathbb{R}^{n \times n \times h}$, with $\mathbf{H}[i,j] \in \mathbb{R}^h$ being the $h$-dimensional embedding of the ordered edge $(i,j)$, a *PPGN* layer updates the graph embedding as follows: Two separate multilayer perceptrons, termed $\text{MLP}_1$ and $\text{MLP}_2$ are applied along the third axis of $\mathbf{H}$ to compute two third-order tensors $\mathbf{M}_1, \mathbf{M}_2 \in \mathbb{R}^{n \times n \times h}$, i.e., for $i,j \in [n]$:

$$\mathbf{M}_1[i,j] = \text{MLP}_1(\mathbf{H}[i,j]) \tag{16}$$
$$\mathbf{M}_2[i,j] = \text{MLP}_2(\mathbf{H}[i,j]). \tag{17}$$

Subsequently, an element-wise matrix multiplication yields $\mathbf{M} \in \mathbb{R}^{n \times n \times h}$. This is computed with $\mathbf{M}[:,:,d] = \mathbf{M}_1[:,:,d] \cdot \mathbf{M}_2[:,:,d]$ for each $d \in [h]$. Equivalently, this operation can be expressed as:

$$\mathbf{M}[i,j] = \sum_{k \in [n]} \mathbf{M}_1[i,k] \odot \mathbf{M}_2[k,j], \tag{18}$$

where $i,j \in [n]$ and $\odot$ denotes the element-wise multiplication. Lastly, the updated graph embedding, represented as $\mathbf{H}'$ is obtained using a third multilayer perceptron, $\text{MLP}_3$, by setting for each $i,j \in [n]$:

$$\mathbf{H}'[i,j] = \text{MLP}_3(\mathbf{H}[i,j] \parallel \mathbf{M}[i,j]), \tag{19}$$

where $\parallel$ signifies the concatenation operation.

### F.2  LOCAL PPGN

In devising our model, we recognized that the graphs of our interest are sparse, a common quality among many real-world graphs. Additionally, our expanded graphs (Definition 1) are locally dense with fully interconnected clusters but globally sparse. Hence, our objective is to construct a model that is expressively local but globally sparse. Our proposed *Local PPGN* which is comparable to the *PPGN* architecture for a fully connected graph and a message passing GNN for a graph devoid of triangles, appears to strike a favorable balance between expressivity and complexity.

The fundamental operation of our *Local PPGN* layer is an edge-wise message passing mechanism. It assumes an underlying directed graph $\overrightarrow{G} = (\mathcal{V}, \overrightarrow{\mathcal{E}})$, where $\mathcal{V}$ is the set of nodes and $\overrightarrow{\mathcal{E}}$ is the set

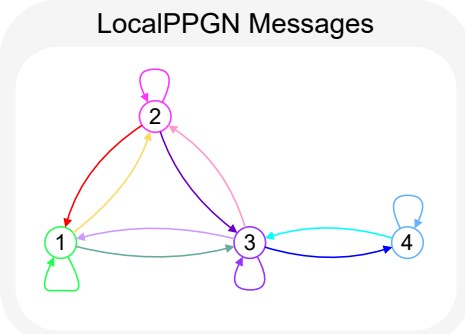

Figure 5: Illustration of a directed graph with self-loops. In the following, we list the update formula for three representative edge embeddings:

$(\boldsymbol{h}')^{(1,1)} = \gamma \left( \boldsymbol{h}^{(1,1)}, \phi \left( \boldsymbol{h}^{(1,1)}, \boldsymbol{h}^{(1,1)} \right) \oplus \left( \boldsymbol{h}^{(1,2)}, \boldsymbol{h}^{(2,1)} \right) \oplus \left( \boldsymbol{h}^{(1,3)}, \boldsymbol{h}^{(3,1)} \right) \right)$

$(\boldsymbol{h}')^{(1,3)} = \gamma \left( \boldsymbol{h}^{(1,3)}, \phi \left( \boldsymbol{h}^{(1,1)}, \boldsymbol{h}^{(1,3)} \right) \oplus \left( \boldsymbol{h}^{(1,2)}, \boldsymbol{h}^{(2,3)} \right) \oplus \left( \boldsymbol{h}^{(1,3)}, \boldsymbol{h}^{(3,1)} \right) \right)$

$(\boldsymbol{h}')^{(3,4)} = \gamma \left( \boldsymbol{h}^{(3,4)}, \phi \left( \boldsymbol{h}^{(3,4)}, \boldsymbol{h}^{(4,4)} \right) \right)$

of directed edges. The emebeddings $\boldsymbol{h}^{(i,j)} \in \mathbb{R}^h$ associated with the edges $(i,j) \in \overrightarrow{\mathcal{E}}$ are updated as follows:

$$(\boldsymbol{h}')^{(i,j)} = \gamma \left( \boldsymbol{h}^{(i,j)}, \bigoplus_{v^{(k)} \in \mathcal{N}^-(v^{(i)}) \cap \mathcal{N}^+(v^{(j)})} \phi \left( \boldsymbol{h}^{(i,k)}, \boldsymbol{h}^{(k,j)} \right) \right). \tag{20}$$

Here, the operator $\bigoplus$ represents a permutation-invariant and differentiable aggregation operation. The functions $\phi$ and $\gamma$ are arbitrary differentiable functions that can be instantiated, for instance, as multi-layer perceptrons. $\mathcal{N}^+(v^{(i)})$ denotes the set of nodes with outgoing edges to $v^{(i)}$, and $\mathcal{N}^-(v^{(i)})$ denotes the set of nodes with incoming edges from $v^{(i)}$.

**Aggregation Sets**  We begin our analysis of this layer by exploring the sets of messages subject to aggregation. For any given node $v^{(i)} \in \mathcal{V}$ and its associated embedding $\boldsymbol{h}^{(i,i)}$, the aggregation set includes the message $\phi(\boldsymbol{h}^{(i,i)}, \boldsymbol{h}^{(i,i)})$. For each neighboring node $v^{(j)}$, the set also contains the message $\phi(\boldsymbol{h}^{(i,j)}, \boldsymbol{h}^{(j,i)})$. For a directed edge $e^{(i,j)} \in \overrightarrow{\mathcal{E}}$, the aggregation set comprises of the message $\phi(\boldsymbol{h}^{(i,i)}, \boldsymbol{h}^{(i,j)})$. Additionally, each triangle $(v^{(i)}, v^{(k)}, v^{(j)})$ present in the graph contributes the message $\phi(\boldsymbol{h}^{(i,k)}, \boldsymbol{h}^{(k,j)})$ to the aggregation set. Figure 5 provides an illustration of the message sets that undergo aggregation for a representative graph.

**Complexity**  The complexity of this layer, from the above analysis, scales linearly with the number of edges and triangles in the graph. For a graph lacking triangles, the complexity therefore scales linearly with the number of edges. Conversely, for a fully connected graph, it scales cubically with the number of nodes.

**Relation to PPGN**  With $\bigoplus$ instantiated as a sum, $\phi$ and $\gamma$ with multi-layer perceptrons as $\phi(\boldsymbol{x}, \boldsymbol{y}) = \mathrm{MLP}_1(\boldsymbol{x}) \odot \mathrm{MLP}_2(\boldsymbol{y})$ and $\gamma(\boldsymbol{x}, \boldsymbol{y}) = \mathrm{MLP}_3(\boldsymbol{x} \parallel \boldsymbol{y})$, we recover the *PPGN* layer as delineated in Section F.1 for a fully connected graph. To discern this, one may compare the scheme 20 with our message passing-like formulation of the *PPGN* layer in Equation 18.

When applied to a general graph, one interpretation of our layer's effects is the application of the *PPGN* layer on each fully connected subgraph within the main graph. This was the initial impetus behind our design choice; we aimed to fuse the expressivity of *PPGN* with the efficiency of message passing GNNs. Given that our graph generation method only introduces local modifications in each step, we theorize that this level of local expressivity is sufficient.

## F.3 ARCHITECTURAL DETAILS

We instantiate our *Local PPGN* model with a succession of the prescribed layers, each responsible for transforming a $d_{\text{hidden}}$-dimensional embedding into another of matching dimensions. For every layer, we set $\bigoplus$ as a sum and normalize by division through the square root of the total message count. We set $\phi(\boldsymbol{x}, \boldsymbol{y}) = \text{MLP}_1(\boldsymbol{x}) \odot \text{MLP}_2(\boldsymbol{y})$, where both $\text{MLP}_1$ and $\text{MLP}_2$ are multi-layer perceptrons, accepting inputs of dimension $d_{\text{hidden}}$ and producing outputs of dimension $d_{\text{PPGN}}$. We define $\gamma(\boldsymbol{x}, \boldsymbol{y}) = \text{MLP}_3(\boldsymbol{x} \parallel \boldsymbol{y})$, where $\text{MLP}_3$ is a multi-layer perceptron with an input dimension of $d_{\text{hidden}} + d_{\text{PPGN}}$ and an output dimension of $d_{\text{hidden}}$.

**MLP Architecture** Within our model, all multi-layer perceptrons consist of two hidden layers. The first has $d_{\text{hidden}}$ neurons, while the second matches the output dimension. Following each hidden layer, a layer normalization (Ba et al., 2016) and a ReLU activation function are applied.

**Input Embeddings** We unify the node, edge and global features (diffusion time step, reduction fraction, target graph size) into a common $d_{\text{emb}}$-dimensional space. Sinusoidal Positional Encodings (Vaswani et al., 2023) are utilized for discrete target graph size embedding, whereas linear projections are used for the remaining features. Global features are replicated for each node and edge in the same graph and are appended to the respective node or edge features. The initial given node embeddings, as outlined in Section 3.5, are also concatenated to the node embeddings. Similarly, given node embeddings corresponding to the edge's endpoints are joined with the edge embedding. Each embedding feature is subject to independent dropout with 0.1 probability before being projected to $d_{\text{hidden}}$ dimensions via a linear layer.

**Output** Final output features for each node or edge are obtained by concatenating the initial embedding and all intermediate embeddings (i.e., outputs of every layer), applying a dropout with 0.1 probability and projecting these features to the desired output dimension using a linear layer.

**SignNet** Our methodology incorporates the *SignNet* model (Lim et al., 2022) for procuring node embeddings from a given graph using the principal spectrum of its Laplacian. Specifically, the $k$ smallest non-zero eigenvalues and the associated eigenvectors of the graph's Laplacian are utilized as input. Each eigenvector concatenated with its corresponding eigenvalue (where eigenvalues are duplicated along the node dimension) is projected into $d_{\text{SignNet}}$ dimensions and subsequently fed into a Graph Isomorphism Network (GIN) (Xu et al., 2019) to generate an embedding. This operation is performed twice for each eigenvector—once for the original eigenvector and once with each dimension negated. The yielded embeddings are then averaged to obtain a sign-invariant embedding. This procedure is repeated for every individual eigenvector, after which all embeddings for each node are concatenated. This concatenated output is subsequently mapped to a $d_{\text{emb}}$-dimensional output using a multi-layer perceptron.

The aforementioned GIN comprises a sequence of message passing layers, where each layer applies a multi-layer perceptron to the summed node embeddings of all adjacent nodes. Analogously to the Local PPGN, all intermediate embeddings from these layers are concatenated, subjected to a dropout operation with a probability of 0.1, and projected to $d_{\text{SignNet}}$ dimensions using a linear layer.

It should be noted that all MLPs within *SignNet* have a hidden dimension of $d_{\text{SignNet}}$, which can diverge from $d_{\text{hidden}}$. Apart from this, all other architectural details remain identical.

**PPGN** We have also designed a version of the *PPGN* model for our baseline one-shot model. Notably, only edge features need to be generated here, which eliminates the need for input node features. Additionally, there are no initial node embeddings, and reduction fraction or target graph size global features. However, outside of these differences, we maintain a consistent architecture with our *Local PPGN* model.

**Implementation Details** We implement the *Local PPGN* and *SignNet* models using the PyTorch Geometric framework (Fey & Lenssen, 2019), the use of sparse graph representations that facilitate scalability to larger graphs. We utilize the standard batching strategy, which involves merging several graphs into a unified disconnected graph, with a batch index to identify the original graph

associated with each node. In contrast, the *PPGN* model is implemented using dense PyTorch operations and representations.

## F.4 GRAPH NEURAL NETWORK ARCHITECTURE ABLATION STUDY

The objective of this study is to demonstrate the superior performance of our proposed *Local PPGN* model, over a conventional node-wise message passing architecture that we refer to as *GINE*. For the purpose of defining this model, let's denote the embedding of node $v^{(i)} \in \mathcal{V}$ as $\boldsymbol{h}^{(i)} \in \mathbb{R}^h$ and the embedding of edge $(i, j) \in \mathcal{E}$ for a graph $G = (\mathcal{V}, \mathcal{E})$ as $\boldsymbol{h}^{(i,j)} \in \mathbb{R}^h$. The updates to these embeddings are performed as follows:

$$\boldsymbol{h}'^{(i)} = \gamma_{\text{node}} \left( \boldsymbol{h}^{(i)}, \bigoplus_{v^{(j)} \in \mathcal{N}(v^{(i)})} \phi \left( \boldsymbol{h}^{(i)}, \boldsymbol{h}^{(i,j)} \right) \right),$$

$$\boldsymbol{h}'^{(i,j)} = \gamma_{\text{edge}} \left( \boldsymbol{h}^{(i,j)}, \boldsymbol{h}'^{(i)}, \boldsymbol{h}'^{(j)} \right).$$

In the above equations, $\mathcal{N}(v^{(i)})$ represents the set of nodes adjacent to $v^{(i)}$. We define $\phi(\boldsymbol{x}, \boldsymbol{e}) = \text{ReLU}(\boldsymbol{x}, \boldsymbol{e})$ and instantiate $\bigoplus$ as a sum, $\gamma_{\text{node}}(\boldsymbol{x}, \boldsymbol{y}) = \text{MLP}_{\text{node}}(\boldsymbol{x} + \boldsymbol{y})$ and $\gamma_{\text{edge}}(\boldsymbol{x}, \boldsymbol{y}, \boldsymbol{z}) = \text{MLP}_{\text{edge}}(\boldsymbol{x} \parallel \boldsymbol{y} \parallel \boldsymbol{z})$, where $\text{MLP}_{\text{node}}$ and $\text{MLP}_{\text{edge}}$ are multi-layer perceptrons. The node update operation is a modified version of the *GIN* layer (Xu et al., 2019), as proposed by Hu et al. (2020). All other aspects of this model, including the MLP design, input embeddings, and output, are kept identical to those in the *Local PPGN* model. The findings from this ablation study are summarized in Table 5. We observe that the *Local PPGN* model consistently outperforms the *GINE* model across all datasets under consideration. The performance improvement is particularly pronounced for planar graphs, underscoring the importance of local expressivity in our model for capturing the structural properties of such graphs.

| | Planar graphs | | Tree graphs | | Proteins |
|---|---|---|---|---|---|
| Model | Ratio ↓ | V.U.N. ↑ | Ratio ↓ | V.U.N. ↑ | Ratio ↓ |
| Local PPGN | 2.1 | 95.0 | 4.0 | 100 | 5.9 |
| GINE | 4.0 | 57.5 | 12.2 | 97.5 | 7.2 |

Table 5: Ablation study of the graph neural network architecture.

## G   END-TO-END TRAINING AND SAMPLING

We provide algorithmic details for the end-to-end training and sampling procedures in Algorithm 6 and Algorithm 7 respectively. Both algorithms assume the deterministic expansion size setting, described in Section 3.7. In the scenario without deterministic expansion size, the only deviation during the training phase is that the model is not conditioned on the reduction fraction. The corresponding sampling procedure in this setting is described in Algorithm 8. All mentioned algorithms rely on the node embedding computation procedure, described in Algorithm 5.

---

**Algorithm 5 Node embedding computation:** This describes the procedure for computing node embeddings for a given graph. Embeddings are computed for the input graph and then replicated according to the cluster size vector.

---

**Parameters:** number of spectral features $k$
**Input:** graph $G = (\mathcal{V}, \mathcal{E})$, spectral feature model $\mathrm{SignNet}_{\boldsymbol{\theta}}$, cluster size vector $\boldsymbol{v}$
**Output:** node embeddings computed for all nodes in $\mathcal{V}$ and replicated according to $\boldsymbol{v}$

1: **function** EMBEDDINGS($G = (\mathcal{V}, \mathcal{E}), \mathrm{SignNet}_{\boldsymbol{\theta}}, \boldsymbol{v}$)
2:      **if** $k = 0$ **then**
3:          $\boldsymbol{H} = [\boldsymbol{h}^{(1)}, \ldots, \boldsymbol{h}^{(|\mathcal{V}|)}] \overset{\text{i.i.d.}}{\sim} N(\boldsymbol{0}, \boldsymbol{I})$               ▷ Sample random embeddings
4:      **else**
5:          **if** $k < |\mathcal{V}|$ **then**
6:              $[\lambda_1, \ldots, \lambda_k], [\boldsymbol{u}_1, \ldots, \boldsymbol{u}_k] \leftarrow \mathrm{EIG}(G)$         ▷ Compute $k$ spectral features
7:          **else**
8:              $[\lambda_1, \ldots, \lambda_{|\mathcal{V}|-1}], [\boldsymbol{u}_1, \ldots, \boldsymbol{u}_{|\mathcal{V}|-1}] \leftarrow \mathrm{EIG}(G)$   ▷ Compute $|\mathcal{V}| - 1$ spectral features
9:              $[\lambda_{|\mathcal{V}|}, \ldots, \lambda_k], [\boldsymbol{u}_{|\mathcal{V}|}, \ldots, \boldsymbol{u}_k] \leftarrow [0, \ldots, 0], [\boldsymbol{0}, \ldots, \boldsymbol{0}]$       ▷ Pad with zeros
10:         **end if**
11:         $\boldsymbol{H} = [\boldsymbol{h}^{(1)}, \ldots, \boldsymbol{h}^{(|\mathcal{V}|)}] \leftarrow \mathrm{SignNet}_{\boldsymbol{\theta}}([\lambda_1, \ldots, \lambda_k], [\boldsymbol{u}_1, \ldots, \boldsymbol{u}_k], G)$
12:     **end if**
13:     $\tilde{G} = (\mathcal{V}^{(1)} \cup \cdots \cup \mathcal{V}^{(p)}, \tilde{\mathcal{E}}) \leftarrow \tilde{G}(G, \boldsymbol{v})$             ▷ Expand as per Definition 4
14:     set $\tilde{\boldsymbol{H}}$ s.t. for all $p \in [|\mathcal{V}|]$: for all $v^{(p_i)} \in \mathcal{V}^{(p)}, \tilde{\boldsymbol{H}}[p_i] = \boldsymbol{H}[p]$     ▷ Replicate embeddings
15:     **return** $\tilde{\boldsymbol{H}}$
16: **end function**

---

---

**Algorithm 6 End-to-end training procedure:** This describes the entire training procedure for our model.

---

**Parameters:** number of spectral features $k$ for node embeddings
**Input:** dataset $\mathcal{D} = \{G^{(1)}, \ldots, G^{(N)}\}$, denoising model $\text{GNN}_{\boldsymbol{\theta}}$, spectral feature model $\text{SignNet}_{\boldsymbol{\theta}}$
**Output:** trained model parameters $\boldsymbol{\theta}$

1: **function** $\text{TRAIN}(\mathcal{D}, \text{GNN}_{\boldsymbol{\theta}}, \text{SignNet}_{\boldsymbol{\theta}})$
2:    **while** not converged **do**
3:       $G \sim \text{Uniform}(\mathcal{D})$                                                        ▷ Sample graph
4:       $(G_0, \ldots, G_L) \leftarrow \text{RNDREDSEQ}(G)$                 ▷ Sample coarsening sequence
5:       $l \sim \text{Uniform}(\{0, \ldots, L\})$                           ▷ Sample level
6:       **if** $l = L$ **then**
7:          $G_{l+1} \leftarrow G_l$
8:          $\boldsymbol{v}_{l+1} \leftarrow \mathbf{1}$
9:          $\boldsymbol{e}_l \leftarrow \varnothing$
10:       **else**
11:          set $\boldsymbol{v}_{l+1}$ as in Eq. 4 and $\boldsymbol{e}_l$ as in Eq. 5, s.t. $G(\tilde{G}(G_{l+1}, \boldsymbol{v}_{l+1}), \boldsymbol{e}_l) = G_l$
12:       **end if**
13:       **if** $l = 0$ **then**
14:          $\boldsymbol{v}_l \leftarrow \mathbf{1}$
15:       **else**
16:          set $\boldsymbol{v}_l$ as in Eq. 4, s.t. the node set of $\tilde{G}(G_l, \boldsymbol{v}_l)$ equals that of $G_{l-1}$
17:       **end if**
18:       $\boldsymbol{H}_l \leftarrow \text{EMBEDDINGS}(G_{l+1}, \text{SignNet}_{\boldsymbol{\theta}}, \boldsymbol{v}_{l+1})$       ▷ Compute node embeddings
19:       $\hat{\rho} \leftarrow 1 - (n_l/n_{l-1})$, with $n_l$ and $n_{l-1}$ being the size of $G_l$ and $G_{l-1}$
20:       $D_\theta \leftarrow \text{GNN}_{\boldsymbol{\theta}}(\cdot, \cdot, \tilde{G}_l, \boldsymbol{H}_l, n_0, \rho)$, where $n_0$ is the size of $G_0$
21:       take gradient descent step on $\nabla_{\boldsymbol{\theta}} \text{DIFFUSIONLOSS}(\boldsymbol{v}_l, \boldsymbol{e}_l, D_\theta)$
22:    **end while**
23:    **return** $\boldsymbol{\theta}$
24: **end function**

---

**Algorithm 7 End-to-end sampling procedure with deterministic expansion size:** This describes the sampling procedure with the deterministic expansion size setting, described in Section 3.7. Note that this assumes that the maximum cluster size is 2, which is the case when using edges as the contraction set family for model training.

---

**Parameters:** reduction fraction range $[\rho_{\min}, \rho_{\max}]$
**Input:** target graph size $N$, denoising model $\text{GNN}_{\boldsymbol{\theta}}$, spectral feature model $\text{SignNet}_{\boldsymbol{\theta}}$
**Output:** sampled graph $G = (\mathcal{V}, \mathcal{E})$ with $|\mathcal{V}| = N$

1: **function** $\text{SAMPLE}(N, \text{GNN}_{\boldsymbol{\theta}}, \text{SignNet}_{\boldsymbol{\theta}})$
2:    $G = (\mathcal{V}, \mathcal{E}) \leftarrow (\{v\}, \emptyset)$                                   ▷ Start with singleton graph
3:    $\boldsymbol{v} \leftarrow [2]$                                                     ▷ Initial cluster size vector
4:    **while** $|\mathcal{V}| < N$ **do**
5:       $\boldsymbol{H} \leftarrow \text{EMBEDDINGS}(G, \text{SignNet}_{\boldsymbol{\theta}}, \boldsymbol{v})$       ▷ Compute node embeddings
6:       $n \leftarrow \|\boldsymbol{v}\|_1$
7:       $\rho \sim \text{Uniform}([\rho_{\min}, \rho_{\max}])$          ▷ random reduction fraction
8:       set $n_+$ s.t. $n_+ = \lceil \rho(n + n_+) \rceil$          ▷ number of nodes to add
9:       $n_+ \leftarrow \min(n_+, N - n)$          ▷ ensure not to exceed target size
10:       $\hat{\rho} \leftarrow 1 - (n/(n + n_+))$          ▷ actual reduction fraction
11:       $D_\theta = \text{GNN}_{\boldsymbol{\theta}}(\cdot, \cdot, \tilde{G}(G, \boldsymbol{v}), \boldsymbol{H}, N, \hat{\rho})$
12:       $(\boldsymbol{v})_0, (\boldsymbol{e})_0 \leftarrow \text{SDESAMPLE}(D_\theta)$       ▷ Sample feature embeddings
13:       set $\boldsymbol{v}$ s.t. for $i \in [n]$: $\boldsymbol{v}[i] = 2$ if $|\{j \in [n] \mid (\boldsymbol{v})_0[j] \geq (\boldsymbol{v})_0[i]\}| \geq n_+$ and $\boldsymbol{v}[i] = 1$ otherwise
14:       $\boldsymbol{e} \leftarrow \text{DISCRETIZE}((\boldsymbol{e})_0)$
15:       $G = (\mathcal{V}, \mathcal{E}) \leftarrow G(\tilde{G}, \boldsymbol{e})$          ▷ Refine as per Definition 2
16:    **end while**
17:    **return** $G$
18: **end function**

**Algorithm 8 End-to-end sampling procedure:** This describes the entire sampling procedure without the deterministic expansion size setting.

---

**Input:** target graph size $N$, denoising model $\text{GNN}_{\boldsymbol{\theta}}$, spectral feature model $\text{SignNet}_{\boldsymbol{\theta}}$
**Output:** sampled graph $G = (\mathcal{V}, \mathcal{E})$

1: **function** SAMPLE($N, \text{GNN}_{\boldsymbol{\theta}}, \text{SignNet}_{\boldsymbol{\theta}}$)
2:     $G = (\mathcal{V}, \mathcal{E}) \leftarrow (\{v\}, \emptyset)$                                    ▷ Start with singleton graph
3:     $\boldsymbol{H} \leftarrow \text{EMBEDDINGS}(G, \text{SignNet}_{\boldsymbol{\theta}}, \mathbf{1})$                          ▷ Initial node embedding
4:     $D_\theta \leftarrow \text{GNN}_{\boldsymbol{\theta}}(\cdot, \cdot, G, \boldsymbol{H}, N)$
5:     $(\boldsymbol{v})_{0, \_} \leftarrow \text{SDESAMPLE}(D_\theta)$
6:     $\boldsymbol{v} \leftarrow \text{DISCRETIZE}((\boldsymbol{v})_0)$                                    ▷ Initial cluster size vector
7:     **while** $|\mathcal{V}| < N$ **do**
8:         $\boldsymbol{H} \leftarrow \text{EMBEDDINGS}(G, \text{SignNet}_{\boldsymbol{\theta}}, \boldsymbol{v})$                       ▷ Compute node embeddings
9:         $D_\theta \leftarrow \text{GNN}_{\boldsymbol{\theta}}(\cdot, \cdot, \tilde{G}(G, \boldsymbol{v}), \boldsymbol{H}, N)$
10:        $(\boldsymbol{v})_0, (\boldsymbol{e})_0 \leftarrow \text{SDESAMPLE}(D_\theta)$                     ▷ Sample feature embeddings
11:        $\boldsymbol{v} \leftarrow \text{DISCRETIZE}((\boldsymbol{v})_0)$
12:        $\boldsymbol{e} \leftarrow \text{DISCRETIZE}((\boldsymbol{e})_0)$
13:        $G = (\mathcal{V}, \mathcal{E}) = G(\tilde{G}, \boldsymbol{e})$                              ▷ Refine as per Definition 2
14:    **end while**
15:    **return** $G$
16: **end function**

---

# H    COMPLEXITY ANALYSIS

In the following, we analyze the asymptotic complexity of our proposed method for generating a graph $G$ with $n$ nodes and $m$ edges. The method involves creating an expansion sequence $(G_L = (\{v\}, \emptyset), G_{L-1}, \ldots, G_0 = G)$ that progressively expands a single node into the graph $G$.

Assuming there exists a positive constant $\epsilon > 0$ such that the number of nodes $n_l$ of $G_l$ satisfies the inequality $n_l \geq (1+\epsilon)n_{l-1}$ for all $0 \leq l < L$ iterates, we can deduce that the length of the expansion sequence does not exceed $L = \lceil \log_{1+\epsilon} n \rceil \in \mathcal{O}(\log n)$. This assumption holds true in the context of deterministic expansion size setting, as delineated in Algorithm 7. In the case of Algorithm 8, although not guaranteed, it is likely to hold as the model is trained to invert coarsening sequences with a minimum reduction fraction of $\rho_{\min}$.

Two observations are made to bound the sizes of the iterates $G_l$ in the sequence. Since expansion can only increase the number of nodes in the graph, no $G_l$ has more than $n$ nodes. For the number of edges, a similar statement cannot be made, as a refinement step can remove arbitrarily many edges. Theoretically, a graph $G_l$ with $l > 0$ could encompass more than $m$ edges. Nevertheless, this scenario is improbable for a trained model - provided sufficient training data and model capacity - as graph coarsening, according to Definition 3, can only decrease the number of edges. Consequently, the coarse graphs used for model training do not contain more edges than the dataset graphs. For the purpose of this analysis, we assume that the number of edges in all $G_l$ is asymptotically bounded by $m$.

Next, we bound the complexity of generating an iterate $G_l$ in the sequence. For $l = L$, this involves instantiating a singleton graph and predicting the expansion vector $\boldsymbol{v}_L$, which can be done in constant time and using constant space. For all other instances where $0 \leq l < L$, given the graph $G_{l+1}$ and the expansion vector $\boldsymbol{v}_{l+1}$, the graph $G_l$ and expansion vector $\boldsymbol{v}_l$ are derived by constructing the expansion $\tilde{G}_l = \tilde{G}(G_{l+1}, \boldsymbol{v}_{l+1})$. This is followed by sampling $\boldsymbol{v}_l$ and $\boldsymbol{e}_l$ and subsequently constructing $G_l = G(\tilde{G}_l, \boldsymbol{e}_l)$. Let $v_{\max}$ denote the maximum cluster size, which is 2 for the edge contraction set family. This allows us to establish an upper bound on the number of edges in the expansion $\tilde{G}_l$. The edge set of $\tilde{G}_l$ comprises, at most, $n_{l+1} \frac{v_{\max}(v_{\max}-1)}{2}$ intracluster edges and a maximum of $m_{l+1} v_{\max}^2$ cluster interconnecting edges. It is reasonable to assume that $v_{\max}$ is constant, as the distribution of expansion sizes can only encompass a constant number of categories. Consequently, no expanded graph will asymptotically exceed $m$ edges. Sampling $\boldsymbol{v}_l$ and $\boldsymbol{e}_l$ is done by querying the denoising model a constant number of times. The complexity for this depends on the underlying graph neural network architecture. For message passing models, this is linear in the number of nodes and edges in the graph. Our *Local PPGN* model also achieves this efficiency if the graph contains at most $\mathcal{O}(m)$ many triangles. Finally, we bound the complexity of obtaining the node embeddings for $G_l$. The first step involves calculating the $k$ main eigenvalues and eigenvectors of the graph Laplacian of $G_{l+1}$. By using the method suggested in Vishnoi (2013), this can be achieved with a complexity of $\tilde{\mathcal{O}}(km_{l+1})$, where $\tilde{\mathcal{O}}$ hides polylogarithmic factors. Computing the node embeddings from this using *SignNet* is done in $\mathcal{O}(km_{l+1})$ time and space. The replication of the embeddings for the expansion $\tilde{G}_l$ is linear relative to the number of nodes in $\tilde{G}_l$. Given that we select $k$ as a constant, the aggregate complexity for computing the node embeddings equates to $\tilde{\mathcal{O}}(n + m)$.

In conclusion, under the stated assumptions, the complexity to generate a graph $G$ with $n$ nodes and $m$ edges is $\tilde{\mathcal{O}}(n + m)$, again hiding polylogarithmic factors.

Figure 6 provides an empirical comparison of our method's runtime efficiency in generating sparse planar graphs against other graph generation models. Our approach achieves subquadratic runtime with respect to the number of nodes, outperforming *DiGress* (Vignac et al., 2023a) and our baseline one-shot method. It operates a constant factor slower than *BwR (EDP-GNN)* (Diamant et al., 2023) and *BiGG* (Dai et al., 2020) but exhibits similar asymptotic scaling. This is remarkable given that our method significantly exceeds these models in terms of sample fidelity, as shown in our experiments.

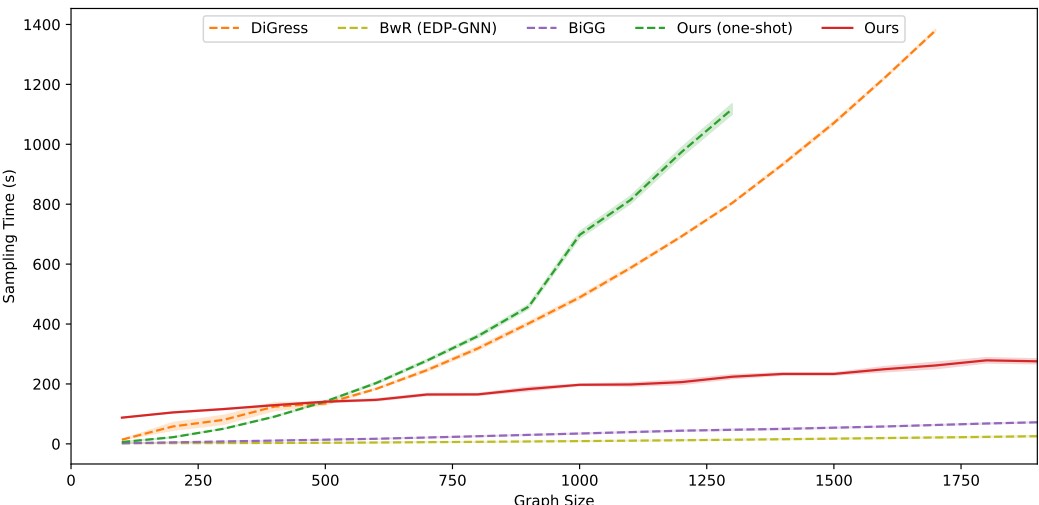

Figure 6: Sampling Efficiency Comparison: The plot illustrates the time required to generate a single planar graph as a function of node count. Mean and standard deviation are calculated over 10 runs. Models with complexity tied to sparsity structure were trained to overfit a single graph, and timing was recorded for generating that graph. Models failing to replicate the target structure, such as those producing disconnected graphs, have been excluded for fairness. Experiments were conducted using a single NVIDIA GeForce RTX A6000 GPU. The reported values encompass all graph sizes that do not surpass the GPU's memory capacity of 40GB.

# I EXPERIMENTAL DETAILS

## I.1 DATASETS

We utilize the following synthetic and real-world datasets for our experimental evaluation:

- **Planar graphs:** This dataset from Martinkus et al. (2022) comprises 200 planar graphs with 64 nodes. These graphs are generated by applying Delaunay triangulation on a set of points placed uniformly at random in the unit square.

- **SBM (Stochastic Block Model) graphs:** We obtain this dataset from Martinkus et al. (2022), consisting of 200 graphs with 2 to 5 communities. Each community contains 20 to 40 nodes, and the probability of an edge between two nodes is 0.3 if they belong to the same community and 0.05 otherwise.

- **Tree graphs:** We generate 200 random trees, using NetworkX (Hagberg et al., 2008), each containing 64 nodes.

- **Protein graphs:** Dobson & Doig (2003) provide a dataset of protein graph representations. In this dataset, each node corresponds to an amino acid and an edge exists between two nodes if the distance between their respective amino acids is less than 6 angstroms.

- **Point cloud graphs:** We adopt the point cloud dataset used in Liao et al. (2020), which consists of 41 point clouds representing household objects (Neumann et al., 2013). As a substantial portion of the graphs are not connected, we only keep the largest connected component of each graph.

For consistency, we employ the train/test split method proposed by Martinkus et al. (2022). Specifically, we allocate 20% of the graphs for testing purposes and then partition the remaining data into 80% for training and 20% for validation. When available we use the exact split by Martinkus et al. (2022). The dataset statistics are presented in Table 6.

| Dataset | Max. nodes | Avg. nodes | Max. edges | Avg. edges | Train | Val. | Test |
| --- | --- | --- | --- | --- | --- | --- | --- |
| Planar | 64 | 64 | 181 | 177 | 128 | 32 | 40 |
| SBM | 187 | 104 | 1129 | 500 | 128 | 32 | 40 |
| Tree | 64 | 64 | 63 | 63 | 128 | 32 | 40 |
| Protein | 500 | 258 | 1575 | 646 | 587 | 147 | 184 |
| Point cloud | 5037 | 1332 | 10886 | 2971 | 26 | 7 | 8 |

Table 6: Dataset statistics.

## I.2 EVALUATION METRICS

We use the same evaluation metrics as Martinkus et al. (2022) to compare the performance of our model with other graph generative models. We report the maximum mean discrepancy (MMD) between the generated and the test graphs for the following graph properties: degree distribution, clustering coefficient, orbit counts, spectrum, and wavelet coefficients. As a reference, we compute these metrics for the training set and report the mean ratio across all of these metrics as a comprehensive indicator of statistical similarity. It is important to note that for the point cloud dataset, given its k-nearest neighbor structure, the degree MMD is identically zero; consequently, it is not incorporated into the mean ratio computation. Analogously, for the tree dataset, both the clustering coefficient and orbit counts are excluded from the ratio computation for the same reasons.

Another set of important metrics are uniqueness and novelty. These metrics quantify the proportion of generated graphs that are not isomorphic to each other (uniqueness) or to any of the training graphs (novelty).

As proposed by Martinkus et al. (2022) for synthetic datasets, we also report the validity score which verifies whether the generated planar graphs retain their planarity, whether the SBM graphs have a high likelihood of being generated under the original SBM parameters, and whether the tree graphs lack cycles.

## I.3 HYPERPARAMETERS AND TRAINING SETUP

We train our model using the Adam optimizer (Kingma & Ba, 2017) and an initial learning rate of $10^{-4}$. A comprehensive summary of the additional hyperparameters employed for our model, as well as the baselines against which we compare, can be found in Table 7. We train all models until there is no further performance improvement on the validation set, with an upper limit of four days. For model selection, we choose the epoch exhibiting the best validation metric, which constituted the fraction of valid, unique, and novel graphs in the case of synthetic datasets, and the mean ratio for real-world datasets.

| Model | Hyperparameter | Planar | SBM | Tree | Protein | Point Cloud | Extrapolation | Interpolation |
|---|---|---|---|---|---|---|---|---|
| | | | | | | Experiment | | |
| Ours | Hidden embedding size ($d_{\text{hidden}}$) | 256 | 256 | 256 | 256 | 256 | 256 | 256 |
| | PPGN embedding size ($d_{\text{PPGN}}$) | 128 | 128 | 128 | 128 | 128 | 128 | 128 |
| | Input embedding size ($d_{\text{emb}}$) | 32 | 32 | 32 | 32 | 32 | 32 | 32 |
| | Number of layers | 10 | 10 | 10 | 10 | 10 | 10 | 10 |
| | Number of denoising steps | 256 | 256 | 256 | 256 | 256 | 256 | 256 |
| | Batch size | 32 | 16 | 32 | 16 | 8 | 32 | 32 |
| | EMA coefficient | 0.99 | 0.999 | 0.99 | 0.9999 | 0.999 | 0.99 | 0.99 |
| | Number of spectral features | 2 | 0 | 2 | 0 | 2 | 2 | 2 |
| | SignNet embedding size ($d_{\text{SignNet}}$) | 128 | — | 128 | — | 128 | 128 | 128 |
| | SignNet number of layers | 5 | — | 5 | — | 5 | 5 | 5 |
| Ours (one-shot) | Hidden embedding size | 256 | 256 | 256 | 256 | — | 256 | 256 |
| | PPGN embedding size ($d_{\text{PPGN}}$) | 64 | 64 | 64 | 64 | — | 64 | 64 |
| | Input embedding size ($d_{\text{emb}}$) | 32 | 32 | 32 | 32 | — | 32 | 32 |
| | Number of layers | 8 | 8 | 8 | 8 | — | 8 | 8 |
| | Number of denoising steps | 256 | 256 | 256 | 256 | — | 256 | 256 |
| | Batch size | 16 | 8 | 16 | 2 | — | 16 | 8 |
| | EMA coefficient | 0.999 | 0.99 | 0.999 | 0.9999 | — | 0.999 | 0.999 |
| GRAN (Liao et al., 2020) | Hidden size | — | — | 128 | — | 256 | 128 | 128 |
| | Embedding size | — | — | 128 | — | 256 | 128 | 128 |
| | Number of layers | — | — | 7 | — | 7 | 7 | 7 |
| | Number of mixtures | — | — | 20 | — | 20 | 20 | 20 |
| | Batch size | — | — | 20 | — | 10 | 20 | 20 |
| DiGress (Vignac et al., 2023a) | Number of layers | 10 | 8 | 10 | 8 | — | 10 | 10 |
| | Number of diffusion steps | 1000 | 1000 | 1000 | 1000 | — | 1000 | 1000 |
| | Batch size | 64 | 12 | 64 | 2 | — | 64 | 12 |
| EDGE (Chen et al., 2023b) | Number of diffusion steps | 128 | 256 | 32 | 256 | 256 | — | — |
| | Batch size | 16 | 8 | 16 | 8 | 2 | — | — |
| BwR (EDP-GNN) (Diamant et al., 2023) | Hidden size | 128 | 128 | 128 | 128 | 128 | — | — |
| | Number of diffusion steps | 200 | 200 | 200 | 200 | 200 | — | — |
| | Batch size | 48 | 48 | 32 | 16 | 2 | — | — |
| BiGG (Dai et al., 2020) | Ordering | DFS | DFS | DFS | DFS | DFS | DFS | DFS |
| | Accumulated gradients | 1 | 1 | 1 | 5 | 15 | 1 | 1 |
| | Batch size | 32 | 32 | 32 | 48 | 2 | 32 | 32 |
| GraphGen (Goyal et al., 2020) | Batch size | 32 | 32 | 32 | 32 | 32 | 32 | 32 |

Table 7: Training hyperparameters for the models used in the experiments. A dash (—) signifies that we did not perform the experiment for the associated dataset, either due to memory restrictions or because the results are sourced from Martinkus et al. (2022). All unspecified hyperparameters default to their standard values. For all additional models, the results are adapted from Martinkus et al. (2022).

## J SAMPLES

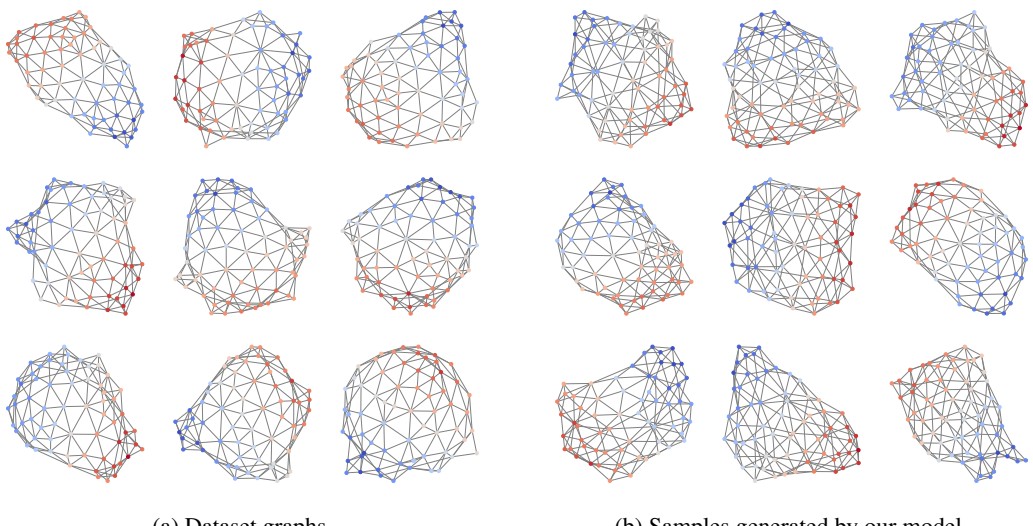

(a) Dataset graphs.                    (b) Samples generated by our model.

Figure 7: Uncurated set of planar graph samples.

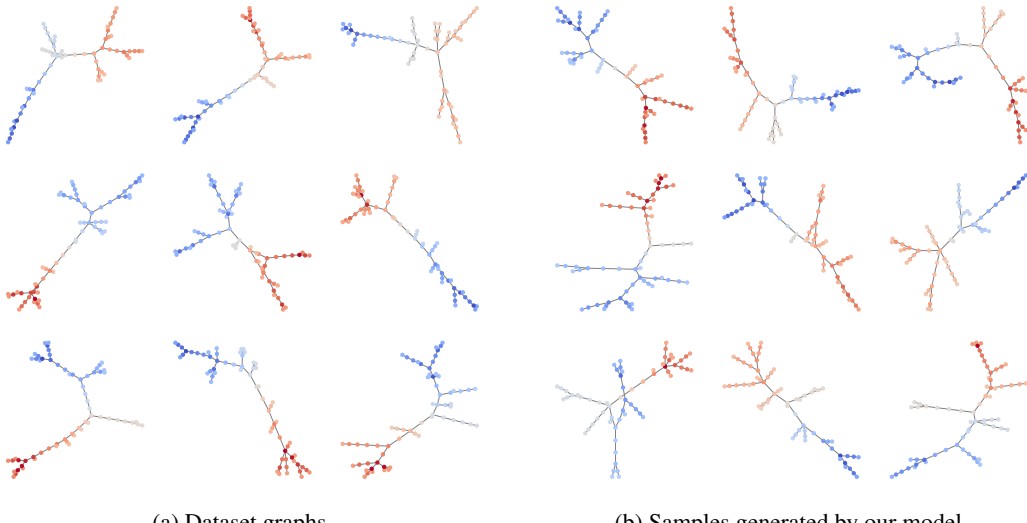

(a) Dataset graphs.                    (b) Samples generated by our model.

Figure 8: Uncurated set of tree graph samples.

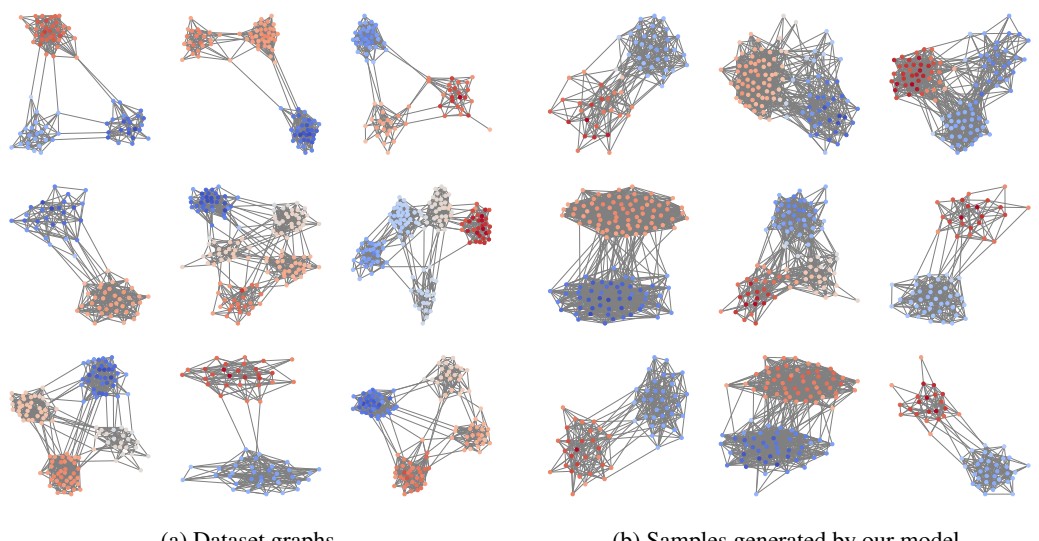

(a) Dataset graphs.                    (b) Samples generated by our model.

Figure 9: Uncurated set of SBM graph samples.

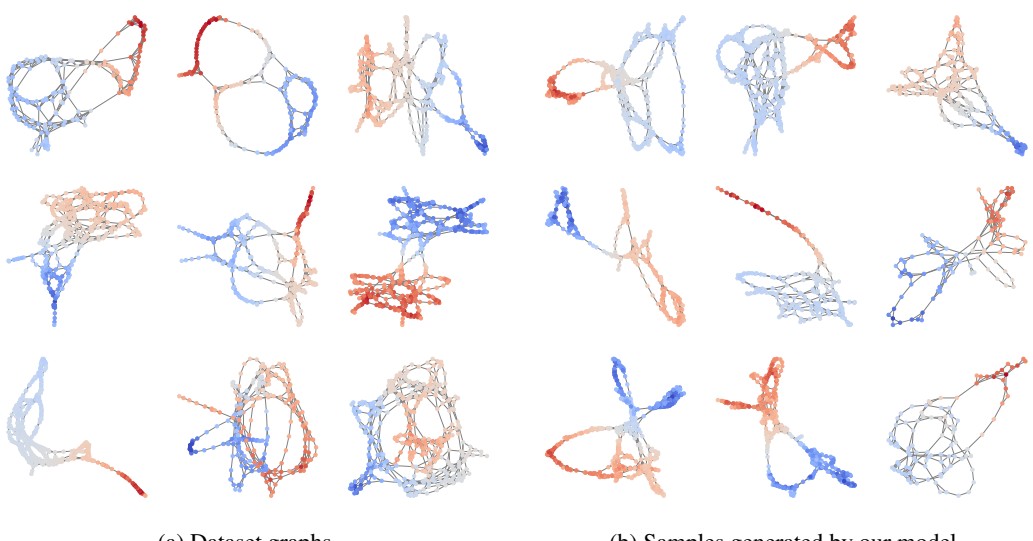

(a) Dataset graphs.                    (b) Samples generated by our model.

Figure 10: Uncurated set of protein graph samples.

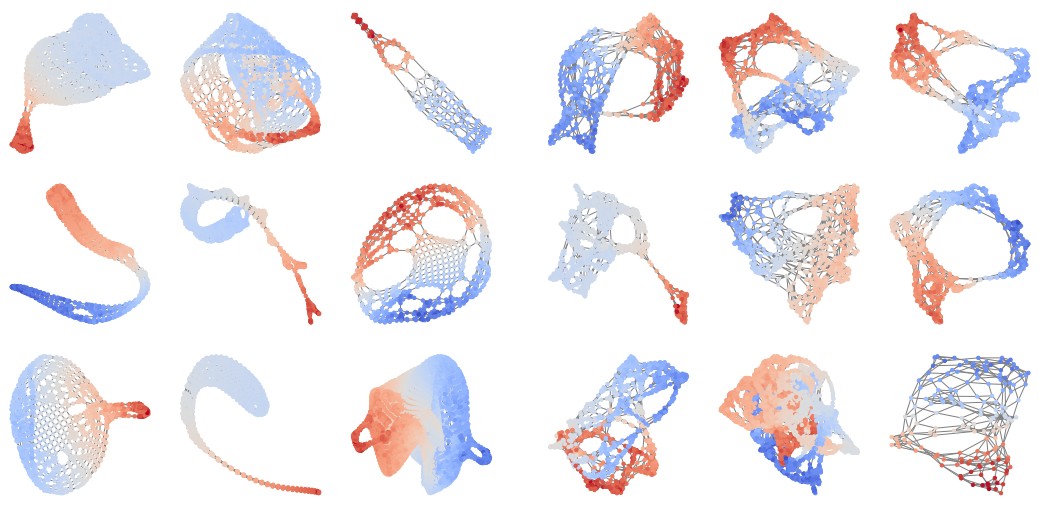

(a) Dataset graphs.      (b) Samples generated by our model.

Figure 11: Uncurated set of point cloud graph samples.

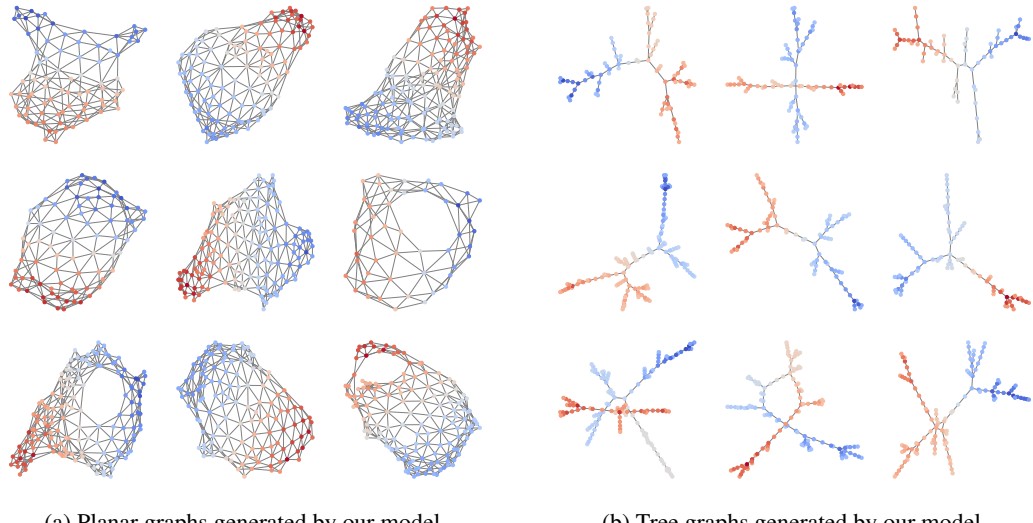

(a) Planar graphs generated by our model.    (b) Tree graphs generated by our model.

Figure 12: Uncurated set of graph samples with 48 to 144 nodes from the extrapolation experiment, surpassing the maximum number of 64 nodes seen during training.

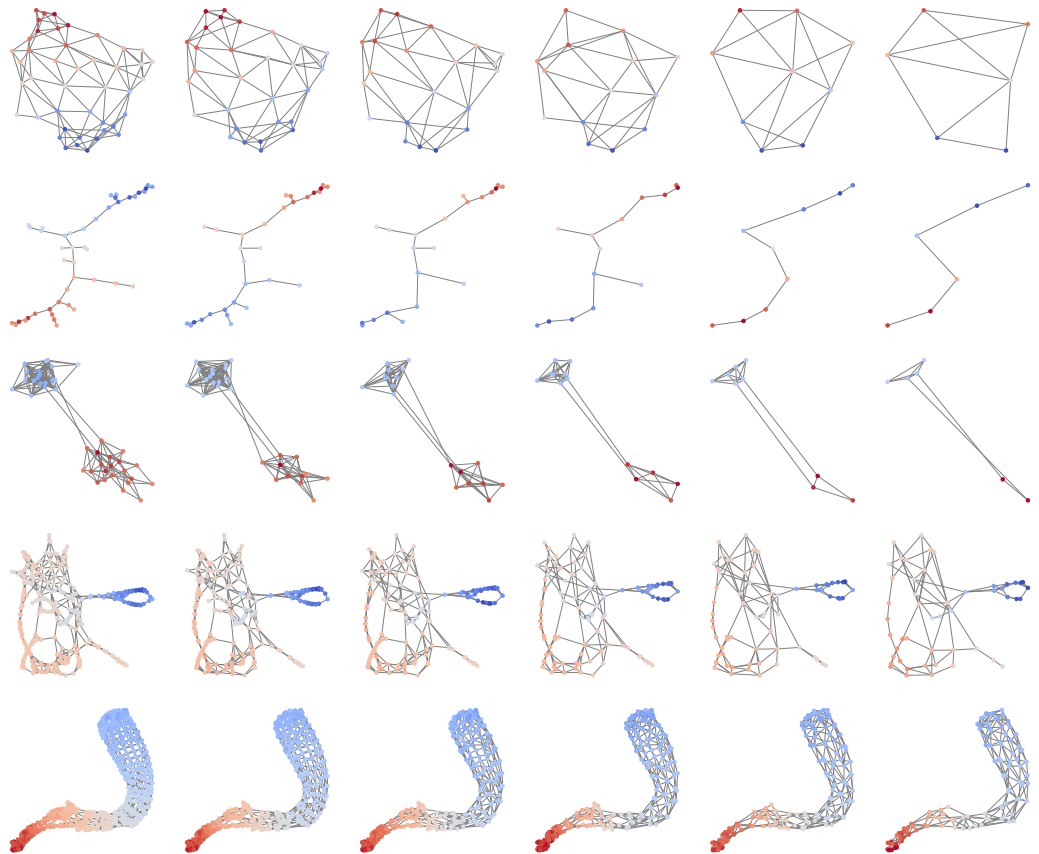

Figure 13: Illustrative example of spectrum preserving coarsening: The first column of each row presents a graph from our datasets. The subsequent columns depict progressive coarsening steps applied to these graphs. Each coarsening step is executed with a consistent reduction fraction of 0.3. The objective of these steps is to maintain spectral characteristics of the original graph throughout the coarsening process, thereby preserving essential structural information.

