# OpenReview forum: "Efficient and Scalable Graph Generation through Iterative Local Expansion"
_ICLR.cc/2024/Conference — ICLR 2024 poster_

### Official Review · Reviewer_38QF · 2023-10-31

**Soundness:** 3 good
**Presentation:** 2 fair
**Contribution:** 2 fair
**Rating:** 6
**Confidence:** 2

**Summary:**

This paper proposes a graph-generative model based on (1) iteratively expanding the coarse graph and (2) spectral conditioning. The authors demonstrate performance improvement over several baselines (GraphRNN GRAN, GDSS, Digress) on planar, SBM, tree, protein, and point cloud graphs.

**Strengths:**

This paper proposes a new method for scalable graph generation, which is an important problem. The idea is sound and the empirical results are solid.

**Weaknesses:**

### Comparison to existing scalable graph generative model (BiGG, GraphGen)

The authors do not compare their results with significant baselines for graph generation. Since the proposed method is designed for scalable graph generation, scalable baselines should be prioritized over others.


[Dai et al., 2020] Scalable Deep Generative Modeling for Sparse Graphs

[Goyal et al., 2022] GraphGen: A Scalable Approach to Domain-agnostic Labeled

[Diamant et al., 2023] Improving Graph Generation by Restricting Graph Bandwidth

### Novelty compared to coarsening-based graph generation.
There exists a work based on generating graphs in a coarse-graining manner. The authors should compare with such a work.

[Guo et al., 2022] An Unpooling Layer for Graph Generation

### Lack of graph visualization and additional metrics for extrapolation
The authors do not visualize the generated graph, which is quite important for verifying the validity of the proposed method, e.g., well-generated graphs are usually connected and indistinguishable from training data from the human eyes.


Furthermore, the authors only check a fraction of valid and unique ratios for interpolation and extrapolation tasks. I suggest reporting additional MMD-based metrics.

### Lack of real-world experiments

All the datasets considered in the experiments are (perhaps arguably) synthetic. For the completeness of the experiment, the authors could consider real-world experiments like molecule generation.

**Questions:**

- I do not understand why the authors argue the V.U.N. metric is the most important metric. Could the authors elaborate on this point?

---

> ### Author Response · Authors · 2023-11-21
> **Answer to reviewer 38QF**
>
> Thank you for the time and effort you have invested in reviewing our work. We appreciate your constructive feedback and have revised the manuscript in response to your comments. In the following, we would like address the concerns and questions you have raised.
>
>
> ## Regarding weaknesses 1 and 2:
> Thank you for highlighting these references. We have expanded the related work section of our manuscript to include a discussion on the following publications:
> - Scalable Deep Generative Modeling for Sparse Graphs
> - Improving Graph Generation by Restricting Graph Bandwidth
> - GraphGen: A Scalable Approach to Domain-agnostic Labeled
> - An Unpooling Layer for Graph Generation
>
> For the first two papers listed, we have carried out experiments using our datasets under consideration and have included the findings in the updated manuscript. We recognize the value of conducting similar experiments for the last two papers mentioned and plan to incorporate these results into the final draft of our paper.
>
>
> ## Regarding weakness 3:
> Graph visualizations have been included in the supplementary material to demonstrate the quality of generation (Figures 7,8,9,10,11,12). As the MMD-based metrics correlate with our reported validity metrics, we have chosen to present only the latter for a more concise presentation.
>
>
> ## Regarding weakness 4:
> Protein and point cloud graphs represent real-world structures by encoding neighborhood information similar to molecular graphs. Therefore, we consider them to be non-synthetic datasets
> The reason we excluded molecular datasets from our evaluation is that they consist of smaller graphs with just a few dozen nodes, which do not align with our method's focus on efficiently generating larger graphs.
> We are optimistic that larger datasets with more nodes per graph will become available, providing further opportunities to evaluate our method on larger graphs.
>
>
> ## Regarding question 1:
> The validity metric is essential as it signifies whether the model has successfully captured the defining attributes of the target graph category. A low score in uniqueness and novelty can be indicative of overfitting, which is particularly important to consider since some models may display low MMD metrics, potentially giving a false impression of high performance. In reality, these models might be replicating the training dataset too closely, failing to generate new, valid structures, which is arguably the main task of generative models. Consequently, we emphasize the V.U.N. (Validity, Uniqueness, Novelty) metric to ensure that our model not only excels statistically but also produces graphs that are novel and characteristic of the intended class. We appreciate you bringing up this issue. To address this, we have included an explanatory note in the manuscript.

---

> > ### Comment · Reviewer_38QF · 2023-11-23
> >
> > I thank the authors for incorporating my comments. I think the paper is better now. I am not entirely satisfied with the rebuttal.
> > However, I believe that the authors are planning to further improve the paper once accepted. Hence, I am raising my score but decreasing my confidence (since my assessment is borderline).
> >
> > Weakness 1 & 2:
> > I think the incorporation of the first two baselines (BiGG and BwR) is great, and implementing two additional ones is of less priority (although it would be better).
> >
> > However, it seems that the performance of BiGG is worse than GRAN for degree, cluster, and orbit MMD metrics for the proteins and the point cloud dataset. This is against the large performance improvement reported in the BiGG paper and it is a bit hard for me to believe that the authors correctly implemented BiGG. Furthermore, in Figure 6 of the appendix, it seems that the proposed method is slower than BiGG.
> >
> > Weakness 3:
> > Thank you for the visualization. I think it would be better to include comparison to the existing baselines too.
> >
> > Weakness 4:
> > I do not think the proteins and the point clouds dataset sufficiently represent real-world problems since they are non-attributed graphs that simplify the real-world attributed objects. The authors could consider polymer datasets similar to HierVAE proposed by Jin et al., 2020.

---

> ### Author Response · Authors · 2023-11-23
> **Further clarification**
>
> Thank you for your response and additional suggestions.
>
> It is true that the performance of BiGG is slightly worse than the performance indicated in the original paper. We utilized the official implementation (https://github.com/google-research/google-research/tree/master/bigg), and for both the protein and point cloud datasets, we applied the same hyperparameters as those documented in the original study. The models were trained until the validation loss stabilized, after which we reported the performance based on the test set. For the final version of the paper we will test what happens when we train them longer. We used a different training split though, as described in the paper, which may explain the difference in performance
> Our method is slower than BiGG by a constant factor. The primary objective of the comparison was to demonstrate the asymptotic scaling behavior to support our theoretical analysis, wherein our method exhibits similarities to BiGG. As we aim to improve diffusion models, which provide state-of-the-art generation quality for smaller graphs, but have very poor asymptotics that prevents their use for larger graphs. We successfully remedy this.
>
> It is indeed correct that the protein and point cloud datasets do not constitute attributed graphs. Our current methodology is not equipped to generate attributed graphs; rather, we focus solely on constructing the topological structure. We appreciate your suggestion to test our model on polymers. This would certainly be a valuable extension.

---

### Official Review · Reviewer_HbqV · 2023-11-01

**Soundness:** 3 good
**Presentation:** 2 fair
**Contribution:** 3 good
**Rating:** 5
**Confidence:** 4

**Summary:**

This paper proposes a novel graph generation method via sequentially expansion. To achieve this, the authors sequentially coarse a graph via graph partition to obtain a sequence of intermediate graphs towards a single-node graph. The graph generation is basically reversing this process, starting from a single node graph and gradually do expansion and refinement towards the original full graph. What's more, each step of the graph refinement for reverting graph coarsen is modeled with denoising diffuison model, while all steps share the same parameters of denoising diffusion model. To establish and concretize this graph coarsen & graph expansion idea, the authors also study and optimize the orderings of coarsen and expansion (spectrum preserving graph coarsen), the underlying diffusion model used (SDE with pre-conditioning and self-conditioning), and also the network architecture (local PPGN to utilize sparsity while retain certain expressivity). The authors demonstrate the improved performance on several (small) plain-graph datasets, and two datasets with larger graphs.

**Strengths:**

1. The designed method is novel. Graph generation via reverting graph coarsen is a promising and interesting approach.
2. The development and concretization of the proposed idea is comprehensive. The authors discuss three additional design space around the idea: the graph coarsening ordering, the underlying diffusion model that utilizes newest improvement from literature, the local PPGN architecture to tradeoff expressivity with computational cost.
3. The presentation in general is easy to follow, with minor problems that I will point out later.
4. The authors designed an experiment of testing extrapolation and interpolation shows the great strength of sequentially expansion over one-shot generation, which is interesting and shows that the designed method is promising.
5. The authors provide detailed configuration of hyperparameters, good for reader to check.
6. I found that the spectrum preserving coarsen and spectrum guided generation are a good design, this constraints all intermediate graphs during the graph expansion process to be inside a family of graphs that having similar spectrum information. This probably leads to the improved extrapolation and interpolation result.

**Weaknesses:**

1. While being comprehensive that touches many design spaces, I personally found that it's a bit harder to know where the improvement of performance comes from. It would be great if the author can provide more detailed ablation study to guide the audience the importance of each design choice.
2. For designed method depends on random orderings of graph coarsen and the reverse ordering of graph expansion. One shortcoming is that the designed method can be sensitive to orderings. The second shortcoming is that the method may needs a longer training period to converge comparing with one-shot generation, and I wish the author to discuss this.
3. To follow up the previous shortcoming, I have noticed that the author only test the designed method over datasets with really limited number of graphs. I concern whether the training time increased due to the sequential coarsen and randomness of coarsen ordering can be too large for a dataset with large number of graphs. For example, typical molecule datasets can have millions of graphs inside. If the designed method has too much improved training cost, then it is not applicable to these real-world cases. Hence, I strongly suggest the author introduce additional datasets instead of just these small datasets. (small = limited number of graphs in the dataset)
4. Decomposition of joint distribution, cannot be sum over all conditional likelihood. Instead should sum over P(G|order) x P(order). Similar issue inside equation 2, equation 3. These parts need a revision.
5. The equation after eq.1 combines two independent distributions together, which uses less information to predict v_l, and can lead to error. This is because that edges being refined can be very important for identifying which node is expected to expand.
6. The equation above (3) is essentially importance sampling, which needs the condition that support of distribution is covered all possible ordering (prob >0 for all orderings).

**Questions:**

1. The definition of edge contraction and neighborhood contraction sets are unclear. How does each element inside the set translate to an order of coarsening?
2. Can you explain how do you achieved batching of graphs when considering randomness of graph coarsening? And how do you batch different steps of graph expansion together? I'm concerning whether this leads to increased computation burden comparing with one-shot generation.
3. When you use eigenvectors, why use SignNet? It is sign invariant representation, should be bad for edge prediction task. It should only be good for graph-level tasks.
4. Can you provide more discussion over the choice of ordering of graph coarsen and graph expansion? From equation 1-3 they seem to play an important role. From your experimental result, how sensitive is the the ordering for generation quality?
5. Can you provide result on molecular datasets? At least a relative small one QM9 should be considered.
6. For one-shot generation, you have set the number of inference steps to be 1000, while for you each intermediate step needs 256 inference steps. Can you provide the total inference steps you used? Are this significantly larger than 1000? In the meantime, a fair comparison should use relatively same number of steps.

---

> ### Author Response · Authors · 2023-11-21
> **Answer 1/2 to reviewer HbqV**
>
> Thank you for your detailed review and constructive feedback. We appreciate your time and effort in evaluating our work.
>
> We would like to address your concerns and questions in the following:
>
> ## Regarding weakness 1:
> In response to the difficulty in discerning the source of performance improvement due to multiple design choices, we have included further commentary in the main text to elucidate the significance of each design choice.
> To avoid misconceptions about the role of spectrum preservation, we will amend the title to "Iterative Local Expansion" in the camera-ready version (per ICLR guidelines). These changes aim to provide a clearer insight into the importance and contribution of individual design elements.
>
>
> ## Regarding weaknesses 2 and 3 as well as questions 4 and 5:
> Thank you for your attention to detail. Let us elaborate on the concerns mentioned, as this provides a good opportunity to clarify some aspects of our method and highlight its strengths.
>
> Utilizing a fixed random coarsening sequence for each graph in the dataset indeed introduces sensitivity to the chosen ordering, which can lead to overfitting when the training set is small. Consequently, we adopted a strategy of sampling different coarsening sequences during model training. This is reflected in the formulation of our method, where importance sampling is employed instead of maximizing the likelihood of a single coarsening sequence, in contrast to other hierarchical graph generation approaches.
>
> Although random coarsening can perform quite well, the spectrum-preserving coarsening proposed reduces the variability of the coarsening sequence and aids in increasing the convergence speed and generative performance.
>
> As stated in the paper, we limited the maximum training time for all experiments to 48 hours. Typically, our method achieves faster convergence compared to baseline one-shot methods due to its efficiency.
>
> Regarding the concern about convergence speed on larger datasets, we experimented by expanding the synthetic planar dataset tenfold and observed no substantial change in the loss convergence rate. However, the generative performance improved, with all generated graphs being valid planar graphs. This improvement is likely because the training data in this case provides better coverage of the ground-truth distribution.
>
> The robust performance on small synthetic datasets underscores the promise of our method. This is particularly noteworthy because several autoregressive methods tend to overfit these datasets and fail to generate valid new graphs.
>
> The reason we excluded molecular datasets from our evaluation is that they consist of smaller graphs with just a few dozen nodes, which do not align with our method's focus on efficiently generating larger graphs.
> We are optimistic that larger datasets with more nodes per graph will become available, providing further opportunities to evaluate our method on larger graphs.
>
>
> ## Regarding weakness 4:
> You are correct in your observation that the marginalization of the likelihood of a graph was incorrectly expressed. This has been corrected in the revised version. Thank you for pointing this out.
>
>
> ## Regarding weakness 5:
> Node expansion indeed depends on the preceding refinement step; that is, $\mathbf{v_l}$ is dependent on $\mathbf{e_l}$. However, as we model a joint distribution $p(\mathbf{v_l}, \mathbf{e_l} | \tilde{G}_l)$, this dependence is captured. The assumption we make is that $\mathbf{v_l}$ is conditionally independent of the expanded graph $\tilde{G}_l$ given the refined graph $G_l$.
>
>
> ## Regarding weakness 6:
> You are correct in your observation that the equality in the equation above (3) is essentially importance sampling and requires $q(\pi|G)>0$ when $p(\pi)>0$. In scenarios where a random cost function is employed during the sampling of coarsening sequences, it holds that the distribution $q$ supports all possible coarsening sequences. With a deterministic cost function, the introduction of the randomization parameter $\lambda$ within the "Randomized Greedy Min-Cost Partitioning" algorithm is intended to ensure a similar support for $q$. However, our primary focus is on employing the variational interpretation to derive a valid lower bound for training purposes rather than evaluating likelihoods. Therefore, to remain general and keep the paper simple, we replace the equality in the equation above (3) with an inequality. This makes our lower bound applicable even in the absence of full support from $q$. Consequently, we have not included a formal argument and refrain from asserting the use of importance sampling in our methodology.

---

> > ### Author Response · Authors · 2023-11-21
> > **Answer 2/2 to reviewer HbqV**
> >
> > ## Regarding question 1:
> > The contraction family limits the potential subgraphs that can be contracted. In each contraction step, a non-overlapping subset of these potential subgraphs is contracted. The specific sets chosen for contraction determine the sequence of contractions, which can vary accordingly.
> >
> >
> > ## Regarding question 2:
> > We implemented our model using the PyTorch Geometric library, following the standard batching convention (https://pytorch-geometric.readthedocs.io/en/latest/advanced/batching.html). This involves merging several graphs into a unified disconnected graph, with a batch index to identify the original graph associated with each node. A clarifying remark has been included in the "Implementation Details" paragraph in Appendix F.3.
> >
> >
> > ## Regarding question 3:
> > Sign invariance can be a beneficial attribute for both graph- and edge-level tasks.
> > For instance, in symmetric graphs, the values of a fixed eigenvector corresponding to automorphic nodes have the same magnitude but are opposite in sign. The sign invariance of SignNet ensures that predictions remain unaffected by arbitrary sign flips of the eigenvectors.
> >
> >
> > ## Regarding question 5:
> > The reason we excluded molecular datasets from our evaluation is that they consist of smaller graphs with just a few dozen nodes, which do not align with our method's focus on efficiently generating larger graphs.
> > We are optimistic that larger datasets with more nodes per graph will become available, providing further opportunities to evaluate our method on larger graphs.
> >
> >
> > ## Regarding question 6:
> > Our iterative expansion and one-shot baseline methods both employ 256 inference steps for denoising diffusion sampling, resulting in a total of 512 model evaluations per sample due to the second-order method used, as specified in the Hyperparameters table. Baseline models follow their original works' specified step counts, such as 1000 for Digress. Typically, further increases in step counts do not yield observed improvements. For reference, please see Figure 2 in https://arxiv.org/pdf/2210.01549.pdf.
> > In the comparison of sampling times, illustrated by a new plot in the revised manuscript, all diffusion methods consistently employ 512 model evaluations.

---

> > > ### Comment · Reviewer_HbqV · 2023-11-22
> > > **Thanks for the detailed response**
> > >
> > > I thank the author's response. Some of my questions are partially cleared. However I still want to see more experimental result regarding ablation study and molecular datasets. At least this method should give comparable result and is runnable for these datasets.

---

> > > > ### Author Response · Authors · 2023-11-23
> > > > **Further clarifications**
> > > >
> > > > Thank you for your response and for taking the time to review our paper once more.
> > > >
> > > > The main focus of the work is to show that the proposed iterative expansion procedure can improve upon scalability and fidelity of graph diffusion models.
> > > >
> > > > ## Ablation study
> > > > We perform the following ablations study.
> > > > 1. The main question is: what is the impact of the proposed iterative expansion? To answer this question, we evaluate a diffusion model without coarsening (Ours (one-shot)) vs our final method (Ours), which is covered by always including the one-shot baseline in all of our experiments. As can be seen, the model using our proposed iterative expansion approach extrapolates much better and is much more efficient than the one-shot baseline. It also almost always improves sample quality.
> > > > 2. A second question is: does the coarsening method used for model training matter? This question is answered by the ablation of the coarsening cost function in Appendix D.1.
> > > > 3. Beside the coarsening scheme, our paper makes another contribution in the form of a Local PPGN architecture. As we aim to build an efficient and scalable generative method, this disqualifies the dense GNN architectures (e.g. graph transformers or original PPGN) for use in the coarsening setup (the point of coarsening is that it allows us to avoid dense edge evaluation). Besides killing scalability, due to memory consumption, using dense GNNs for coarsening model would make computation time prohibitively long. So the appropriate comparison is standard message-passing GNNs, out of which GINE is maximally expressive. We perform this ablation in Appendix F.4 and see that Local PPGN is superior.
> > > >
> > > >
> > > > ## Molecule generation
> > > > As the goal of our contribution is to improve the scalability and quality of graph generation, we have focused our effort only on the graph structure and our method, currently does not support the generation of attributed graphs. An extension is possible but not straightforward, as it requires defining how features are coarsened.
> > > > While there are methods for featured graph coarsening (see [Kumar et al., 2023](https://proceedings.mlr.press/v202/kumar23a/kumar23a.pdf)), integrating these into our system is deferred to future work.
> > > > Due to these limitations, we cannot compare our molecule generation capabilities with other methods and report standard metrics like validity at this time.
> > > > Additionally, molecule generation does not fit the scalability setting. Consequently, other scalable graph generation baselines have also refrained from testing with this type of data ([BiGG](https://arxiv.org/pdf/2006.15502.pdf), [HiGen](https://arxiv.org/pdf/2305.19337.pdf), [HiGGs](https://arxiv.org/pdf/2306.11412.pdf)).
> > > > What would be a much more fitting and interesting application in our opinion, going towards strong real-world applications is 3D protein structure generation, akin to [RFDiffusion](https://www.biorxiv.org/content/10.1101/2022.12.09.519842v1), where we do have large graph structures, inaccessible to common graph diffusion generative models. This would be very promising for our model, as it already displays much better performance than existing models on the current protein dataset, with medium sized proteins.

---

### Official Review · Reviewer_mLXw · 2023-11-03

**Soundness:** 3 good
**Presentation:** 2 fair
**Contribution:** 3 good
**Rating:** 8
**Confidence:** 3

**Summary:**

The paper describes a generative method for graphs. The generative procedure is cast in terms of reversing a graph hierarchical coarsening algorithm that groups nodes of the original graph in a hierarchical way to approximately preserve the spectra of the graph.
The authors propose to model the un-coarsening as a sequence of expansion - in which each node is expanded to a clique of a sampled number of nodes - and refinement steps. This later is tasked to remove edges from the expanded graph.
Both expansion and refinement are modeled in a Bayesian framework, where the joint distribution of the number of expansion nodes (for the next step) and edges to keep are conditioned on the previous graph in the hierarchical un-coarsening process.
Sampling from this distribution is performed through a denoising diffusion algorithm.

**Strengths:**

This paper provides a rigorous formulation (even if not always easily accessible) of graph generation in terms of graph uncoarsening in a probabilistic framework. The experimental section shows that the proposed method achieves results comparable (or better) to state-of-the-art graph generative models in standard datasets and shows that the method is able to deal with datasets with graphs of thousands of nodes.

**Weaknesses:**

The main limitation of the problem is that it is not easy to follow. It somehow lacks a level of abstraction that allows the reader to understand the individual parts of the method in an “intuitive” way before going into the details. You really need to carefully read and understand each bit of the 3.1 and 3.2 to understand where the method is headed.

The second weakness is the description and discussion of the local spectral preservation. Being the main focus of the title, I was expecting this to be one of the most important contributions of the paper. Unfortunately, it is briefly described in one paragraph in the main paper, and a small ablation experiment on this component is only provided in the appendix.
In general, I think that an ablation study on the most important design (e.g.  Spectral Conditioning) choices performed by the authors is missing and does not allow one to understand their impact.

I also missed the presence of a brief paragraph positioning the paper with respect of the SOTA and highlighting its contributions.

**Questions:**

There are a few things that are not clear in the method description:
- at the end of page 4 it is mentioned that you have to model a probability distribution over node and edge features. Why features? aren’t v and e the number of nodes distribution and edge selection vector?
- In the probabilistic model description, isn’t a graph G unequivocally defined by an expansion sequence? That is, given an expansion sequence, shouldn’t be the probability of a graph G either 1 or 0?
- I would include some high-level details on how you model p(v_t,e_t | G_t) as a diffusion process. For instance, how are v and e modeled? Is e an adjacency matrix? Do you need to solve an inverse diffusion process for each expansion step?
- Related to the previous point, if you need to sample through the denoising diffusion process, how does this impact the sampling time?

---

> ### Author Response · Authors · 2023-11-21
> **Answer to reviewer mLXw**
>
> Thank you for your detailed review and constructive feedback.
> Although our paper provides a high-level summary and visual aids, we understand that technical details can be difficult to follow. We hope that the reworked algorithms in the appendix will help clarify the method.
>
> In response to your comments on local spectral preservation, we concur that it does not represent the primary focus of our work. As such, we will amend the title to "Iterative Local Expansion" in the camera-ready version (per ICLR guidelines), which more accurately encapsulates the essence of our method. While spectrum-preserving graph coarsening does contribute to the method's enhanced performance, it is not a critical component.
>
> Furthermore, we have expanded the related work section to provide a clearer context within the current state-of-the-art landscape, thereby highlighting our contributions more effectively. We have also enriched the experimental section with additional comparisons to underscore the method's efficacy and impact.
>
>
> In the following, we would like to give answers to your specific questions:
>
> ## Regarding questions 1 and 3:
> The expansion vector records the size of the expansion cluster for each node in the graph $\tilde{G_t}$, whereas the refinement vector specifies whether each edge of this graph should be kept or removed in the refinement step. Consequently, we represent these two vectors as node and edge features of the graph $\tilde{G_t}$. This is why we refer to them as features. We have clarified this in the text. Thank you for pointing this out.
>
>
> ## Regarding question 2:
> The marginalization of the likelihood of a graph has been corrected in the revised version. You are correct in your observation a graph is defined by an expansion sequence resulting in it. Thank you for raising this concern.
>
>
> ## Regarding question 4:
> We discretize the diffusion process into 256 steps. The second-order method employed necessitates two gradient evaluations per step, leading to a total of 512 model evaluations for each sample. Although this procedure multiplies the sampling time by a constant factor, it does not alter the asymptotic complexity.
> A comment stating this has been added to Section 3.3 in the revised manuscript.

---

> > ### Comment · Reviewer_mLXw · 2023-11-22
> >
> > Thanks for the clarifications.
> > I will have a closer look at the modification to the manuscript in the next few days.

---

### Official Review · Reviewer_sNQQ · 2023-11-03

**Soundness:** 2 fair
**Presentation:** 2 fair
**Contribution:** 2 fair
**Rating:** 5
**Confidence:** 4

**Summary:**

In this study, the authors introduce a graph generative model that iteratively expands and refine a coarse graph into a larger graph. This model learns to invert a graph coarsening process by training a denoising diffusion model. To overcome the complexity problem of the available GNN models, a local GNN model based on the PPGN model is developed that concentrates on the local sub-graph expansion.

**Strengths:**

1) Multi-scale graph generation from a coarse graph to fine details, adopted in this work, is important to learn the complex graphs which can also capture global structure and local details of the graphs. It is used as a key part to design efficient and scalable graph generative models.

2) It designed an efficient local version of the PPGN model for parameterizing the local expansion and refinement model.

**Weaknesses:**

1) While the paper introduces a hierarchical and multiscale graph generation approach, it fails to acknowledge and compare it to previous methods [1, 2, 6] with similar objectives. For instance, HiGen [2] employs a hierarchical generation scheme, iteratively un-coarsing graphs and generating subgraphs in a coarse-to-fine manner with state-of-the-art quality. A hierarchical normalizing flow model for molecular graphs is also ignored [1], where new molecular graphs are generated by inverting edge contraction coarsening and duplicating node attributes after each expansion.

2) The method maintains a  constant  number of eigenvalues $k$ (principal Laplacian spectrum) in the expansion process, which means that a fixed global structure of rank $k$ is used while the graph size is growing. On the other hand, since a local PPGN focuses on the local dense subgraph, therefore such *spectrum preserving local expansion* method may not capture some details that lie in the gap between the principal Laplacian spectrum (fixed global structure of the graph) and the local subgraphs. This can limit the model expressivity for larger graphs where this gap is larger.

3) It lacks a comprehensive complexity analysis, despite claiming sub-quadratic complexity. It is essential to compare the sampling times to recent scalable graph generation models. Furthermore, there is no visualization of the generated graphs, which is crucial for assessing the quality of the model's output.

4) The paper falls short in evaluating the generation quality by not comparing it to recent methods such as HiGen [2], GraphARM [3], and EDGE [4]. In particular, the performance on the Tree dataset is only compared against two benchmarks. It is also recommended to include the evaluation of the proposed model using random GNNs based on [5] as an alternative evaluation method to provide a more comprehensive assessment.

5) The paper's presentation requires improvement to enhance its readability.

-  a) Notations can be confusing, with some instances of the same notation used for different terms. For instance, distinguishing the expansion vector $\mathbf{v}$ from nodes using different notation and handling variations in symbols such as $\mathit{\Pi}$ and $\Pi$ would be beneficial. Additionally, the dual use of $\theta$ for representing both the distribution over coarsening sequences and model parameters in section 3.3 may lead to confusion.

- b) Contraction family and contraction function are both denoted by $\mathcal{F}(G)$ while the family need not be a function of input graphs.

- c) The paper should be improved for self-containment and clarity. For example, “we found that the formulation proposed by Song et al. (2021), supplemented with enhancements from Karras et al. (2022), yielded the most promising results, similarly to the approach used in Yan et al. (2023)” is very vague and one need to read the appendix. Also, a significant portion of the crucial model components is pushed to the appendix.

- d) The paper employs the term "autoregressive" to describe the model, which might cause confusion. In section 2, it's mentioned that "one-shot generative models have generally outperformed autoregressive ones." To avoid confusion, it's advisable to use more accurate terminology, such as "recursive" or "iterative," to distinguish the model's nature.

- e) The factorization mentioned at the bottom of page 4, based on the chain rule, should explicitly note the assumption of a Markov property in the expansion sequence, i.e., $P(G_{l-1} | G_l, G_{l+1}, …, G_{L} ) = P(G_{l-1} | G_l)$. This additional clarification will help readers better understand the underlying assumptions.

[1] Maksim Kuznetsov and Daniil Polykovskiy. Molgrow: A graph normalizing flow for hierarchical molecular generation. In Proceedings of the AAAI Conference on Artificial Intelligence, volume 35, pp. 8226–8234, 2021.

[2] M. Karami, “HiGen: Hierarchical Graph Generative Networks”, arXiv preprint arxiv:2305.19337

[3] Lingkai Kong, Jiaming Cui, Haotian Sun, Yuchen Zhuang, B. Aditya Prakash, and Chao Zhang. Autoregressive diffusion model for graph generation, 2023.

[4] Xiaohui Chen, Jiaxing He, Xu Han, and Li-Ping Liu. Efficient and degree-guided graph generation via discrete diffusion modeling. arXiv preprint arXiv:2305.04111, 2023.

[5] Rylee Thompson, Boris Knyazev, Elahe Ghalebi, Jungtaek Kim, and Graham W Taylor. On evaluation metrics for graph generative models. arXiv preprint arXiv:2201.09871, 2022.

[6] Hamed Shirzad, Hossein Hajimirsadeghi, Amir H Abdi, and Greg Mori. Td-gen: Graph generation using tree decomposition. In International Conference on Artificial Intelligence and Statistics, pp. 5518–5537. PMLR, 2022.

**Questions:**

1) It seems that the model is using two iterative processes: an outer process to expand a graph ($G_l -> G_{l-1}$) and an inner process that uses a denoising diffusion model at each $ l $? . It would be helpful to provide a clear explanation of these processes, especially in the context of using the one-shot method.

2) Should the likelihood of a graph G presented before eq (1) be marginalized as:
$P(G) = \sum P(G|\omega)p(\omega)$?

Also, it seems more suitable to rather define it as marginalized over the coarsening sequences
$P(G) = \sum P(G|\pi)p(\pi)$
Also, in the first equation in variational interpretation, it is defined as marginalization over the a contraction family not all possible contraction sequences so it might be corrected by the upper bound:
$P(G) \ge \sum_{\pi in \Pi_{F}} P(G|\pi)p(\pi)$

3) Why do we need a multilevel coarsening scheme that ensures preservation of the Laplacian spectrum? Is this particular property a key to the model, or could other coarsening methods also be utilized which preserve global structures by explicitly preserving approximate Laplacian spectrum?

4) In the footnote of page 5, an example is given for the expansion sequences that are not the reverse of any coarsening sequence. Are there other cases, or can the refinement step be restricted to ensure the refined graph remains connected?

5) It would be valuable to elaborate on why this  added source of randomness in the expansion is beneficial to the model's performance.


6) What was the range of $L$ in each dataset and what family of contractions did you use? Did you limit $L$? In the case $L=1$, is this method (one-shot) equivalent to one of the already available non-multiscale diffusion models in the literature such as Yan et al. (2023)?

7) How the denoising process is focused on the local subgraphs details, did you only add noise to a subset of the edges at each diffusion step?

---

> ### Author Response · Authors · 2023-11-21
> **Answer 1/2 to reviewer sNQQ**
>
> Thank you for the time and effort you have invested in reviewing our work. We appreciate your constructive feedback and have revised the manuscript in response to your comments.
> In the following, we would like to address the concerns and questions you have raised.
>
>
> ## Regarding weaknesseses 1 and 4:
> We appreciate your attention to these references.
> We have revised our related work section to include a discussion of the cited publications, specifically addressing the following studies:
> - Molgrow: A graph normalizing flow for hierarchical molecular generation
> - HiGen: Hierarchical Graph Generative Networks
> - Autoregressive diffusion model for graph generation
> - Efficient and degree-guided graph generation via discrete diffusion modeling
> - Td-gen: Graph generation using tree decomposition
>
> For the work "Efficient and degree-guided graph generation via discrete diffusion modeling", we have conducted experiments on the considered datasets and report the results in the revised manuscript. We recognize the value of conducting additional experiments on the other mentioned works and plan to incorporate these in the final version of the paper. Note that following the suggestion of Reviewer 38QF, we also conducted experiments with two other methods: "Scalable Deep Generative Modeling for Sparse Graphs" and "Improving Graph Generation by Restricting Graph Bandwidth".
>
> In terms of employing random GNNs for evaluation, we recognize its potential significance. However, due to time limitations, we are currently unable to expand our experimental evaluations to include this measure. Nevertheless, we believe that the structural properties captured by the random GNN metric would likely show a correlation with the MMD values we have reported.
>
>
> ## Regarding weaknesses 2:
> We recognize your concerns regarding the potential limitations of our method, which arise from keeping the number of eigenvalues $k$ constant during the expansion process. However, it appears there might be a misunderstanding about how our model operates and the role of spectral conditioning.
>
> To clarify, our Local PPGN model functions across the entire graph rather than on isolated local subgraphs, enabling it to capture global structural characteristics. The term "local" indicates that the model resembles the PPGN model within local dense subgraphs, which results in enhanced discriminative power in these areas compared to message-passing models. The expressiveness of our GNN at the global scale is at least the same as usual message-passing models. Spectral conditioning is not crucial for capturing global structural attributes; rather, its aim is to enhance the model's performance by providing useful node embeddings. We have revised the "Spectral Conditioning" section to more clearly explain this aspect.
>
>
> ## Regarding weaknesses 3:
> In response to your comment regarding the absence of a comprehensive complexity analysis, we have incorporated a new section in Appendix H. We clearly outline the assumptions that allow for the achievement of subquadratic complexity in relation to the number of nodes. This section also includes a new chart that illustrates empirical sampling speeds, benchmarked against alternative methods.
>
>
> ## Regarding weaknesses 5:
> Thank you for your detailed suggestions. We have revised the manuscript to address the concerns you highlighted.
> In particular, we use "q" to denote the probability distribution over coarsening sequences, to avoid confusion with the model parameters.
>
> The revised version avoids the term "autoregressive" to describe the method.
> Furthermore, it explicitly notes the Markov property assumption in the factorization of the joint distribution.
>
> We have elucidated the representation of the expansion and refinement vectors $\mathbf{v}_l,\mathbf{e}_l$ as node and edge features, respectively, of the expanded graph $\tilde{G}_l$, in response to the query raised by reviewer mLXw. Consequently, we maintain that boldface notation for these vectors is appropriate and have chosen to retain it.
>
> The symbol $\mathcal{F}(G)$ is exclusively used to denote the set of candidate contraction sets for a given graph $G$. While we recognize that this may not be the most conventional notation for contraction families, its meaning is unambiguous within the context provided.
>
> We acknowledge that a considerable amount of methodological detail has been relegated to the appendix. Given the constraints of page limits, our intention was to succinctly present the core aspects of our methodology in the main text while offering comprehensive elaboration in the appendix.

---

> ### Author Response · Authors · 2023-11-21
> **Answer 2/2 to reviewer sNQQ**
>
> ## Regarding question 1:
> Your observation regarding the model's use of two iterative processes is accurate. For each refinement and expansion step, we sample corresponding expansion and refinement vectors using denoising diffusion, which is an iterative process. We have updated Section 3.3 on denoising diffusion in the main text to more explicitly point this out.
>
>
> ## Regarding question 2:
> You are correct in your observation that the marginalization of the likelihood of a graph was incorrectly expressed.
> Furthermore, you are correct that the first equation under the variational interpretation section should be an inequality rather than an equality.
> Both of these errors have been corrected in the revised version - thank you for pointing them out.
>
>
> ## Regarding question 3
> Our method supports any coarsening scheme that operates by contracting connected subgraphs.
>
> Coarsening that preserves the Laplacian spectrum ensures that the resulting graphs share structural similarities with the original dataset graphs, which may ease the learning process. Our ablative studies indicate that such preservation can enhance model performance, when compared to random coarsening, albeit modestly. We invite further research to investigate alternative coarsening strategies within our framework.
>
>
> ## Regarding question 4
> Indeed, there are additional scenarios. Consider the expansion of a triangular graph into a fully connected four-node graph, followed by an edge removal (refinement) that results in a line graph. This refined graph cannot revert to the original triangular structure through coarsening by contracting any connected subgraph. Generally, expansion and refinement may not be reversible when the refinement step eliminates edges in such a way that nodes originating from the same parent node no longer share a common candidate contraction set.
>
>
> ## Regarding question 5
> The incorporation of randomness in the expansion process primarily serves to mitigate overfitting, which is particularly advantageous for synthetic datasets comprising only 128 training samples. We added a comment to the "Perturbed Expansion" section to clarify this point.
>
>
> ## Regarding question 6
> As documented in the paper, edge contractions were utilized for all experiments. This approach limits the graph's maximum expansion factor to 2, which in turn means that L is upper bounded by \log_2(n), where n represents the number of nodes in the graph. For the purpose of model training, coarsening sequences were constructed, with each step reducing the graph's size by a random fraction ranging from 10\% to 30\% (Details are provided in Algorithm 1). For instance, for the synthetic graphs of size 64, L ranges from 12 to 22.
>
> Addressing your question about the scenario in which the initial node is directly expanded to the target graph using a one-shot approach: Our method employs the same denoising diffusion framework as "SwinGNN" by Yan et al. However, while they apply a non-permutation invariant transformer architecture to graph edges, we utilize the introduced Local PPGN model—which, in this case of a dense graph, would match the original PPGN model by Haggai et al.
>
> ## Regarding question 7:
> In the refinement step, we selectively remove some edges from the expanded graph $\tilde{G}$. To do this, we sample a refinement vector using denoising diffusion. This vector is represented as edge features of the expanded graph $\tilde{G}$, where each edge feature indicates whether the corresponding edge should be kept or removed during the refinement step. Consequently, noise is added to all edge features of the expanded graph during training.

---

> > ### Comment · Reviewer_sNQQ · 2023-11-23
> >
> > Thank you for your detailed response and clarifications. To help the reviewers' assessment, it would be beneficial to upload a diff version that highlights the changes with a different color in the supplement.
> > I have a few minor concerns and suggestions:
> > 1. **Appendix, GRAPH NEURAL NETWORK ARCHITECTURE ABLATION STUDY:**
> >    a. In comparing the performance of local PPGN against GINE, did you add spectral embedding as node embedding for GINE in a manner similar to local PPGN?
> >    b. Considering the intention to update the paper title, is it necessary to refer to the GNN method as "local PPGN" when, as indicated in the appendix and your response, it functions as an "edge-wise message-passing" GNN? While it is mentioned that it is "comparable to the PPGN architecture for a fully connected graph," this still requires further elaboration for readers.
> >    c. Since there are other edge-wise message-passing GNNs in the literature, consider running an ablation study to compare against them. Additionally, a comparison with more powerful GNNs like transformer-based methods like GT (graph transformer models, as used in DiGress) or hybrid methods such as GraphGPS could further highlight the unique contributions of local PPGN.
> > 2. **Results of EDGE:**
> >    The results for EDGE appear to be anomalous, showing very poor performance across all datasets. This raises skepticism about whether EDGE was trained and evaluated accurately during the rebuttal period.
> >
> > The updates have notably improved the presentation and clarified certain aspects. I plan to read the revised version thoroughly and will consider updating my score after further discussion with other reviewers.

---

> > > ### Author Response · Authors · 2023-11-23
> > > **Further clarifications**
> > >
> > > We appreciate your response and thank you for taking the time to evaluate our paper again.
> > >
> > > A diff-pdf will be added to the supplementary materials.
> > >
> > > 1a. In our comparison of Local PPGN with GINE, we only altered the GNN architecture while maintaining the rest of the model unchanged, such as using SignNet for spectral embedding. We trust that the revised algorithmic description, where node embedding computation is outlined as a separate subroutine (Algorithm 5), clarifies this point.
> > >
> > > 1b. The purpose of revising the title is to characterize our method as "Iterative Local Expansion" rather than "Spectrum Preserving Local Expansion". We believe this more accurately reflects the essence of the method since the spectrum-preserving aspect is not an essential component of the technique but rather a means to enhance the model's performance and accelerate convergence by restricting the coarsened graphs to a class with spectral properties similar to the original graph. We have named our GNN architecture "Local PPGN" because it operates solely on the sparse graph structure, not on the dense adjacency matrix. A more comprehensive explanation of how it relates to PPGN can be found in Appendix F.2 under the "Relation to PPGN" section.
> > >
> > > 1c. As our goal is to develop an efficient and scalable generative method, this approach excludes the use of a dense graph transformer as employed in DiGress. While there are other edge-wise message-passing GNNs documented in the literature, we believe that GINE, being maximally expressive, serves as a representative benchmark for comparison.
> > >
> > > 2 . We did our best to provide accurate results for all new methods (including [EDGE](https://arxiv.org/pdf/2305.04111.pdf)) in the limited time of the rebuttal period. The datasets used in the EDGE paper and our own work are distinct, which poses a challenge for direct evaluation.
> > > We believe that our method should be generalized to generate features as well, and then be compared to EDGE on the same datasets. However, such an extension is beyond the scope of this contribution.
> > > Similarly, [Improving Graph Generation by Restricting Graph Bandwidth](https://arxiv.org/pdf/2301.10857.pdf) performs poorly in our evaluation. Once again, comparison with the original work is challenging, as the datasets and training setup used differ from ours. It is possible that this method is designed for smaller graphs as protein graphs in this study are confined to a maximum of 125 nodes.

---

### Author Response · Authors · 2023-11-21
**Answer to all reviewers**

Dear Reviewers,

Thank you for your constructive feedback. We have thoroughly revised the manuscript in response to your comments.

First, we acknowledge the error in our original treatment of the marginalization of the likelihood of a graph given an expansion sequence, $p(G | \varpi)$. As correctly pointed out, this probability is binary— 1 if the expansion sequence results in the graph, or 0 otherwise. Our intention was to express the likelihood of a graph as the aggregate of probabilities across all possible expansion sequences that culminate in that graph. This error has been amended in the revised manuscript, with no alterations to the fundamental method.

In addition, we have revised certain sections of the text to better address your comments and improve clarity.
We have also reworked the algorithms presented in Appendix G to enhance readability and facilitate a better understanding of our methods. We trust that these revisions will clear up any potential confusion regarding our approach.

Furthermore, we have expanded the related work section to more clearly position our research with respect to the current state-of-the-art, thereby more effectively showcasing our contributions.

The experimental section now includes 3 additional comparisons to underscore the method's effectiveness and importance:
- Efficient and degree-guided graph generation via discrete diffusion modeling
- Scalable Deep Generative Modeling for Sparse Graphs
- Improving Graph Generation by Restricting Graph Bandwidth

To support our claims about efficiency, we have added an analysis of sampling complexity in Appendix H. A new chart showing empirical sampling speeds (Figure 6), benchmarked against other methods, has also been included. It can be empyrically seen that our method results in sub-quadratic complexity to generate sparse graphs.

We have added visual examples of samples created by our model to the appendix.

Finally, we have decided to change the title from 'Efficient and Scalable Graph Generation by Spectrum Preserving Local Expansion' to 'Efficient and Scalable Graph Generation through Iterative Local Expansion' in the camera-ready version, per ICLR guidelines. This change more accurately represents the main elements of our method. As some reviewers pointed out, spectrum-preserving coarsening is not the central focus of our work, nor is it essential to the method; it is rather a way to improve convergence speed and sample quality. We believe this new title better matches our findings and hope you agree with this change.

To compensate for the additional content, we have succinctly rephrased parts of the text to respect the page limit. No content has been removed.

We appreciate your feedback and look forward to your final evaluation.

---

### Meta-Review · Area_Chair_xZHi · 2023-12-07

**Metareview:**

Paper studies a graph generation framework which sequentially “uncoarsens” graphs. This is done via a creating synthetic sequence of graphs which is created by coarsening training dataset graphs while preserving the original spectrum. Their framework is trained to predict “next graph” in this sequence in reverse. Paper creatively use multiple known ML techniques to create this sound framework which achieves competent generative performance in small datasets with large graphs. If published at ICLR, authors are requested to add the promised baseline experiments with fair hyperparameter tuning in the next revision.

**Justification For Why Not Higher Score:**

Missing experiments

**Justification For Why Not Lower Score:**

Novel ideas and good results for an important problem

---

### Decision · Program_Chairs · 2024-01-16

Accept (poster)